# Modulation of the tick gut milieu by a secreted tick protein favors *Borrelia burgdorferi* colonization

Sukanya Narasimhan[1], Tim J. Schuijt[2], Nabil M. Abraham[1,3], Nallakkandi Rajeevan[4,5], Jeroen Coumou[2], Morven Graham[6], Andrew Robson[7], Ming-Jie Wu[1], Sirlei Daffre[8], Joppe W. Hovius[2] & Erol Fikrig [1,3]

The Lyme disease agent, *Borrelia burgdorferi*, colonizes the gut of the tick *Ixodes scapularis*, which transmits the pathogen to vertebrate hosts including humans. Here we show that *B. burgdorferi* colonization increases the expression of several tick gut genes including *pixr*, encoding a secreted gut protein with a Reeler domain. RNA interference-mediated silencing of *pixr*, or immunity against PIXR in mice, impairs the ability of *B. burgdorferi* to colonize the tick gut. PIXR inhibits bacterial biofilm formation in vitro and in vivo. Abrogation of PIXR function in vivo results in alterations in the gut microbiome, metabolome and immune responses. These alterations influence the spirochete entering the tick gut in multiple ways. PIXR abrogation also impairs larval molting, indicative of its role in tick biology. This study highlights the role of the tick gut in actively managing its microbiome, and how this impacts *B. burgdorferi* colonization of its arthropod vector.

---

[1] Department of Internal Medicine, Yale University School of Medicine, New Haven, CT 06420, USA. [2] Department of Internal Medicine, Division of Infectious Diseases, Center for Experimental and Molecular Medicine, Academic Medical Center, University of Amsterdam, Amsterdam, AZ 1105, The Netherlands. [3] Howard Hughes Medical Institute, Chevy Chase, MD 20815, USA. [4] Yale Centre for Medical Informatics, 300 George Street, New Haven, CT 06511, USA. [5] Clinical Epidemiology Research Centre, VA Cooperative Studies Program, West Haven, CT 06516, USA. [6] Yale Centre for Cellular and Molecular Imaging, 333 Cedar Street, New Haven, CT 06510, USA. [7] Program in Vertebrate Developmental Biology, Departments of Pediatrics and Genetics, Yale University School of Medicine, New Haven, CT 06420, USA. [8] Departamento de Parasitologia, Universidade de São Paulo, São Paulo 05508-900, Brazil. Sukanya Narasimhan and Tim J. Schuijt contributed equally to this work. Correspondence and requests for materials should be addressed to S.N. (email: sukanya.narasimhan@yale.edu) or to E.F. (email: erol.fikrig@yale.edu)

*xodes scapularis* is an important vector of the Lyme disease agent, *Borrelia burgdorferi*[1]. *B. burgdorferi* is maintained in a mammal-tick infectious cycle and engages in intimate interactions with *I. scapularis* during its colonization of the tick gut and subsequent transmission to the vertebrate host[2]. These events, concomitant with tick feeding, are accompanied by dramatic changes in the expression profiles of *Borrelia*[3] and tick genes[4] and are pivotal drivers of colonization and transmission.

Using a subtractive hybridization approach[5] to analyze changes that occur in the tick gut transcriptome during *Borrelia* growth and migration from the gut, we find several genes upregulated in *Borrelia*-infected tick guts, including a gene that encodes a secreted protein with a Reeler domain[6], henceforth referred to as Protein of *I. scapularis* with a Reeler domain (PIXR). PIXR functions to limit bacterial biofilm formation and consequently helps maintain bacterial homeostasis in the tick gut. PIXR abrogation results in increased biofilms in the tick gut, and alterations in the gut microbiome and metabolome. We provide evidence that suggests that these changes in the gut microbiome composition and architecture may influence *B. burgdorferi* in diverse ways including by altering the tick immune responses. The current study also suggests that changes in the gut microbiome impact *B. burgdorferi* gene expression and raises the possibility that tick gut microbiota and their metabolites might serve as additional cues that signal changes in the spirochete transcriptome during spirochete colonization of the tick gut.

## Results

**Subtractive hybridization of *I. scapularis* guts.** Subtractive hybridization analysis was performed to explore the influence of *B. burgdorferi* on tick gut gene expression. Only clones that provided high-quality sequences were included, and clones that appeared at least five times were considered specific and are shown in Supplementary Table 1. Clones that code for mitochondrial cytochrome C oxidase subunits I and III appeared at least 10 times. Cytochrome C oxidase, a key mitochondrial enzyme in the respiratory chain, was shown to be important for the transmission of *Anaplasma marginale*[7]. Several genes coding for the short-chain dehydrogenase family of proteins involved in the metabolism of amino acids, lipids or sugars[8] and for proteins involved in defense responses[9], bloodmeal digestion[10], detoxification of endo and xenobiotics[11], complement inhibition[12], lipid transport[13], actin assembly, cell signaling[14–16] and for proteins paralogous to secreted salivary proteins including Salp26A[17] were also upregulated by the presence of *Borrelia* in the tick gut (Supplementary Table 1). Several genes encoded hypothetical proteins or proteins with unknown functions. Two putative immune response genes, Scapularisin-5 (ISCW005926), and a Reeler domain containing protein, henceforth referred to as Protein of *Ixodes scapularis* with a Reeler domain (PIXR, GenBank accession code KY629420) were also increased during *Borrelia* growth. Scapularisin belongs to the defensin family of antibacterial peptides and in silico assessment of the *I. scapularis* genome has revealed 25 Scapularisins[18]. A synthetic peptide based on one of the Scapularisins (Scapularisin-20) was shown to have potent bactericidal activity against gram-positive bacteria[18].

We focused on PIXR, potentially representing a novel immune response of *I. scapularis*. In silico analysis of PIXR revealed homology with Noduler (GenBank ABG72705), a protein involved in immunity in *Antheraea mylitta*[19] (Fig. 1a). Noduler has been shown to be upregulated upon bacterial infection[20], and was shown to bind several microorganisms in order to control and clear infection[19]. The mature PIXR protein consists of a Reeler domain (Fig. 1b), a ~140 amino acids residues long domain located at the N-terminus of a variety of secreted and

surface proteins[6] including Noduler and the other homologues[19]. Homologs of PIXR are found in *Aedes aegypti* (GenBank EAT35239), *Anopheles gambiae* (GenBank XP_315224), *Culex quinquefasciatus* (GenBank XP_001849402) and *Pediculus humanus* (GenBank XP_002426361) with 46–52% similarity (Fig. 1a). PIXR homologs are also represented in other *Ixodid* ticks including *I. ricinus*, the vector of *B. burgdorferi sensu lato* in Northern Europe that elaborates several PIXR-like homologs in hemocytes[21] (shown in Fig. 1a is one homolog, GenBank JAP74644), *Amblyomma sculptum* (GenBank JAU03031) and *A. aureolatum* (GenBank JAT91861) (Fig. 1a). BLASTP analysis of PIXR against the *I. scapularis* genome database revealed six additional paralogs of PIXR (Fig. 1c). Of the seven PIXR paralogs, only PIXR, PIXR2 and PIXR3 contain a predicted signal cleavage site (Fig. 1b, c), indicating that these are likely secreted proteins. One additional *I. scapularis* paralog of the PIXR family (ISCW003629) lacks the Reeler domain and not included in this analysis.

**Characterization of PIXR.** Quantitative PCR with reverse transcription (qRT-PCR) assessment of *pixr* expression showed that *pixr* transcripts are induced upon feeding in nymphal and larval ticks (Fig. 2a, b). Consistent with the subtractive hybridization data (Supplementary Table 1), expression of *pixr* was increased in *B. burgdorferi*-infected nymphal guts compared to uninfected nymphs (Fig. 2a). *pixr* levels were comparable; however, in larvae that fed on clean or *B. burgdorferi*-infected mice (Fig. 2b). Transcript levels of *pixr2* were also induced upon feeding and increased in *B. burgdorferi*-infected nymphal guts when compared to uninfected nymphal guts (Supplementary Fig. 1A). Unlike *pixr*, *pixr2* transcripts were comparably expressed in unfed and fed larvae and in larvae that fed on clean or *B. burgdorferi*-infected mice. While *pixr3*, *pixr4* and *pixr5* transcripts were detected in both nymphal and larval stages the transcript levels were not significantly altered in the presence of *B. burgdorferi* in larval and nymphal stage (Supplementary Fig. 1B–D). Expressions of *pixr6* and *pixr7* could not be detected in larval or nymphal stages. Since *pixr* was identified in our subtractive hybridization screen we assessed the functional role of PIXR in further detail.

PIXR was expressed in a *Drosophila* expression system, and recombinant PIXR (rPIXR) was purified using a Ni-NTA Superflow column (Fig. 2c) and used to generate polyclonal antibodies in rabbit. Western blot analysis of gut and salivary gland extracts (SGEs) of 72 h fed nymphal ticks showed a ~15 kDa doublet band that reacted with the polyclonal anti-PIXR serum in gut extracts, but not in SGEs (Fig. 2d and Supplementary Fig. 8A). Western blot analysis of protein extracts of unfed and fed larval and nymphal ticks showed that PIXR was induced upon feeding (Fig. 2e and Supplementary Fig. 8B)) and a doublet band at ~15 kDa was observed in both larval and nymphal stages. Liquid chromatography/tandem mass spectrometry (LC-MS/MS) analysis of the doublet band revealed peptides corresponding to PIXR (RVVVQWLAPEDRS; SEASPFMGFLIKA) and PIXR 2 (KAFDENEKDVGSFRA; RLVQDTEDYKPGDVITVTLSS). PIXR has a predicted molecular mass of 15.408 kDa and two predicted *O*-glycosylation and one predicted ε-amino group glycation sites, although rPIXR did not reveal presence of glycosylations as judged by Periodic acid-Schiff-base glycoprotein staining (Supplementary Fig. 1E). Full-length mRNA transcript of *pixr 2* (GenBank accession code KY865270) was amplified by PCR from fed larvae and fed nymphal guts and sequenced. The full-length mRNA is 99% identical to the nucleotide sequence of ISCW017733 and encodes a mature protein with a predicted molecular mass of ~15.4 kDa,

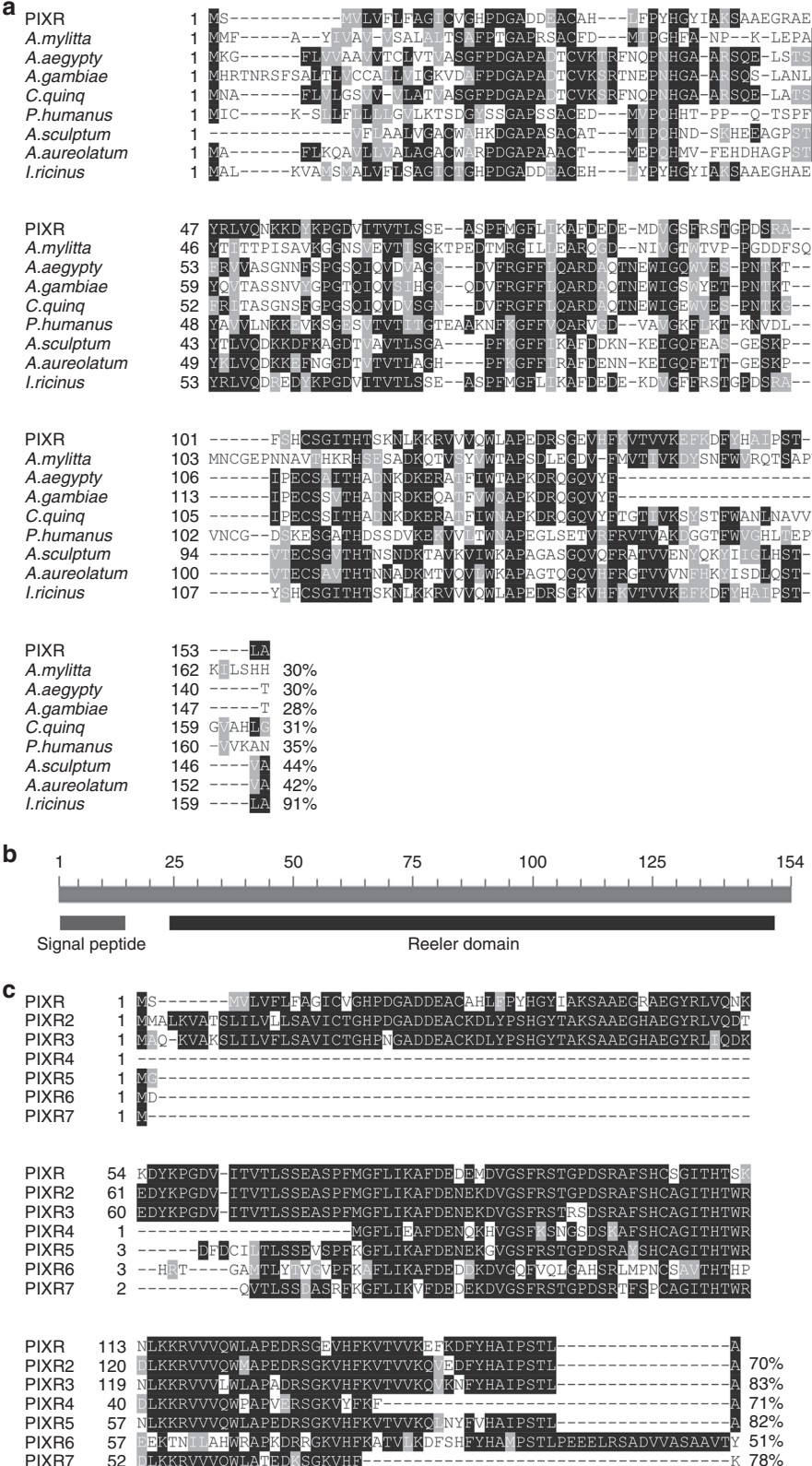

**Fig. 1** PIXR and its homologs in arthropods. **a** Multiple sequence alignment of *Ixodes scapularis* PIXR with the amino acid sequences of several homologs identified in *Antheraea mylitta* (GenBank accession code ABG72705), *Aedes aegypti* (GenBank EAT35239), *Anopheles gambiae* (GenBank XP_315224), *Culex quinquefasciatus* (GenBank XP_001849402) and *Pediculus humanus* (GenBank XP_002426361), *Amblyomma sculptum* (GenBank JAU03031), *Amb. aureolatum* (GenBank JAT91861) and *I. ricinus* (GenBank JAP74644). Amino acids in *white* on a *black background* are identical; residues in *white* on a *gray* background are similar. Similarities and identities to PIXR are presented at the end of the amino acid sequences. **b** Schematic overview of PIXR showing that the mature protein for the greater part consists of a Reeler domain. Scale bar represents number of amino acids. **c** Multiple sequence alignment of *I. scapularis* PIXR with the amino acid sequences of paralogs identified in the *I. scapularis* genome

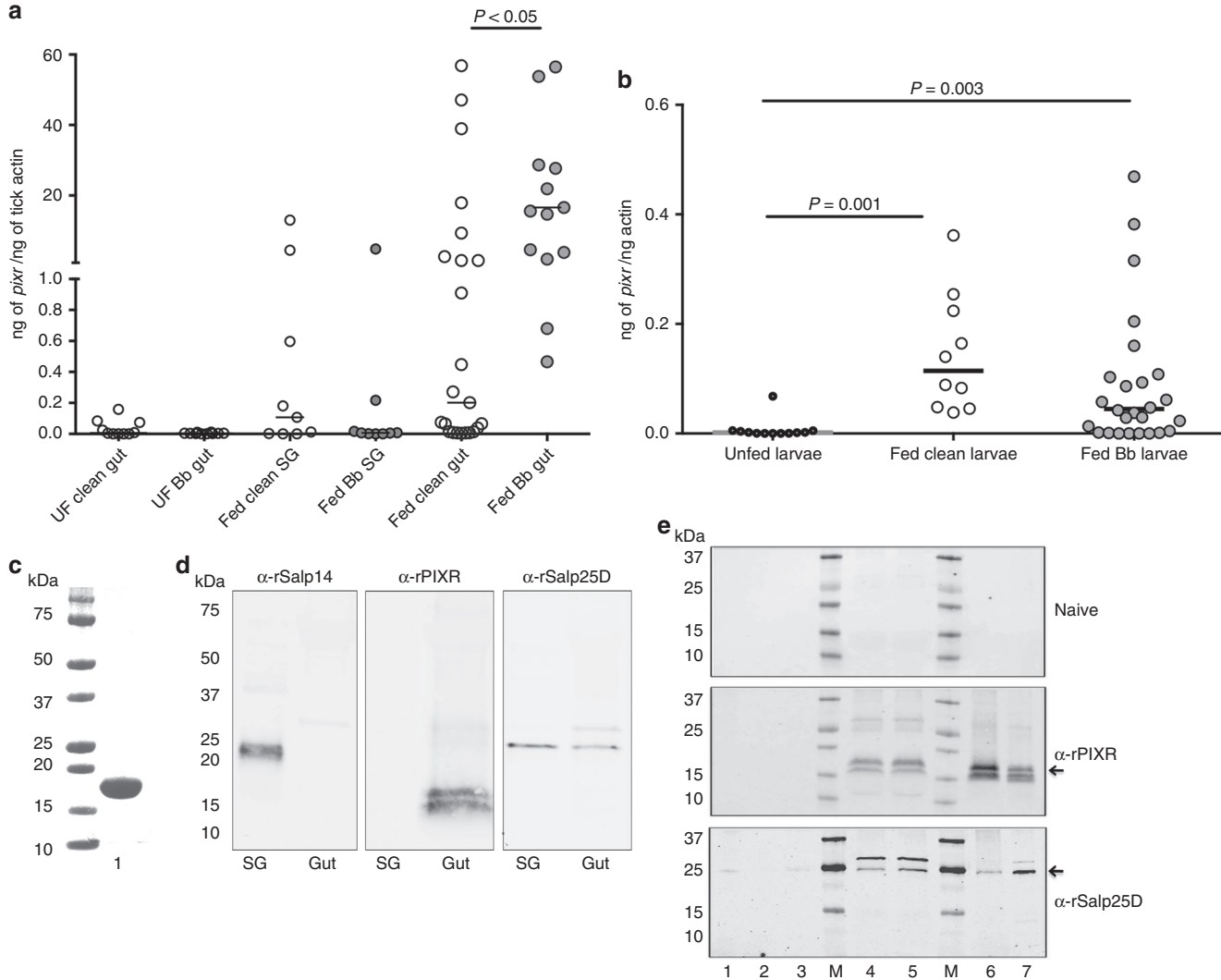

**Fig. 2** PIXR is induced upon feeding in *I. scapularis* nymphs and larvae. Quantitative RT-PCR assessment of *pixr* transcript levels in: **a** unfed clean (UF clean gut) and unfed *B. burgdorferi*-infected (UF Bb gut) gut, fed salivary glands from clean (Fed clean SG) and *B. burgdorferi*-infected nymphs salivary glands (Fed Bb SG), fed guts of clean (Fed clean guts) and *Borrelia*-infected (Fed Bb gut) nymphs. Each data point represents a pool of three nymphal salivary glands or guts. **b** clean unfed larvae (Unfed larvae) and larvae fed on clean (Fed clean larvae) or *B. burgdorferi*-infected mice (Fed Bb larvae). Each data point represents a pool of five fed larvae; Horizontal bar represents the median. Mean values significantly different in a non-parametric Mann–Whitney test ($P < 0.05$) indicated. **c** Coomassie blue staining of purified recombinant PIXR (rPIXR) produced in a *Drosophila* expression system, *lane 1*. Protein markers shown to the left (Precision Plus Biorad protein markers). **d** Western blot of protein extracts of fed clean nymphal salivary glands (SG) and gut (Gut) probed with polyclonal rabbit anti rSalp14 sera, anti-rPIXR or anti rSalp25D sera. **e** Replicate western blots of protein extracts of unfed larvae lane 1; unfed *B. burgdorferi*-infected nymphal gut, lane 2; unfed clean nymphal gut, lane 3; larvae fed on *B. burgdorferi*-infected, lane 4; or on clean mice, lane 5; guts of fed *B. burgdorferi*-infected nymphs lane 6; or fed clean nymphs lane 7; probed with polyclonal naive rabbit sera, or polyclonal anti-rPIXR or anti rSalp25D rabbit sera. Protein markers (Precision Plus Biorad protein markers) are shown in lane M and their cognate molecular weights denoted to the *left*. *Arrows* indicate PIXR and Salp25D-specific protein bands

and has five predicted O-glycosylation sites and four predicted ε-amino group glycation sites. The absence of cross-reacting PIXR3 (predicted molecular mass of ~15.4) in the doublet band suggests low protein expression. No cross-reacting bands corresponding to PIXR 4 and 5 were observed at the predicted sizes of ~7.2 and 10.9 kDa, respectively. Consistent with the qRT-PCR data, PIXR/PIXR2 protein levels were comparable in larvae that fed on uninfected mice or *B. burgdorferi*-infected mice, and increased (~1.5-fold) in *B. burgdorferi*-infected nymphal gut when compared to that in uninfected nymphal guts. Although *pixr2* transcripts were observed in unfed larvae (Supplementary Fig. 2A), no specific cross-reactivity to anti-PIXR antibodies were observed.

**PIXR knockdown impairs *B. burgdorferi* colonization.** On the basis of the homology of PIXR with arthropod putative defense response proteins (Fig. 1a), we hypothesized that RNA interference (RNAi) mediated silencing of PIXR would abrogate a PIXR-driven defense response against *Borrelia* and result in increased spirochetal load in tick guts during transmission to the host and colonization of the tick. We first determined the physiological role of PIXR in the context of *B. burgdorferi* colonization. To this end, ds (double-stranded) *pixr* RNA was injected into the guts of pathogen-free *I. scapularis* nymphs by anal pore injection and ticks allowed to feed to repletion on *B. burgdorferi*-infected mice. The engorgement weights of ds *pixr* RNA -injected nymphs and control dsRNA-injected nymphs were comparable

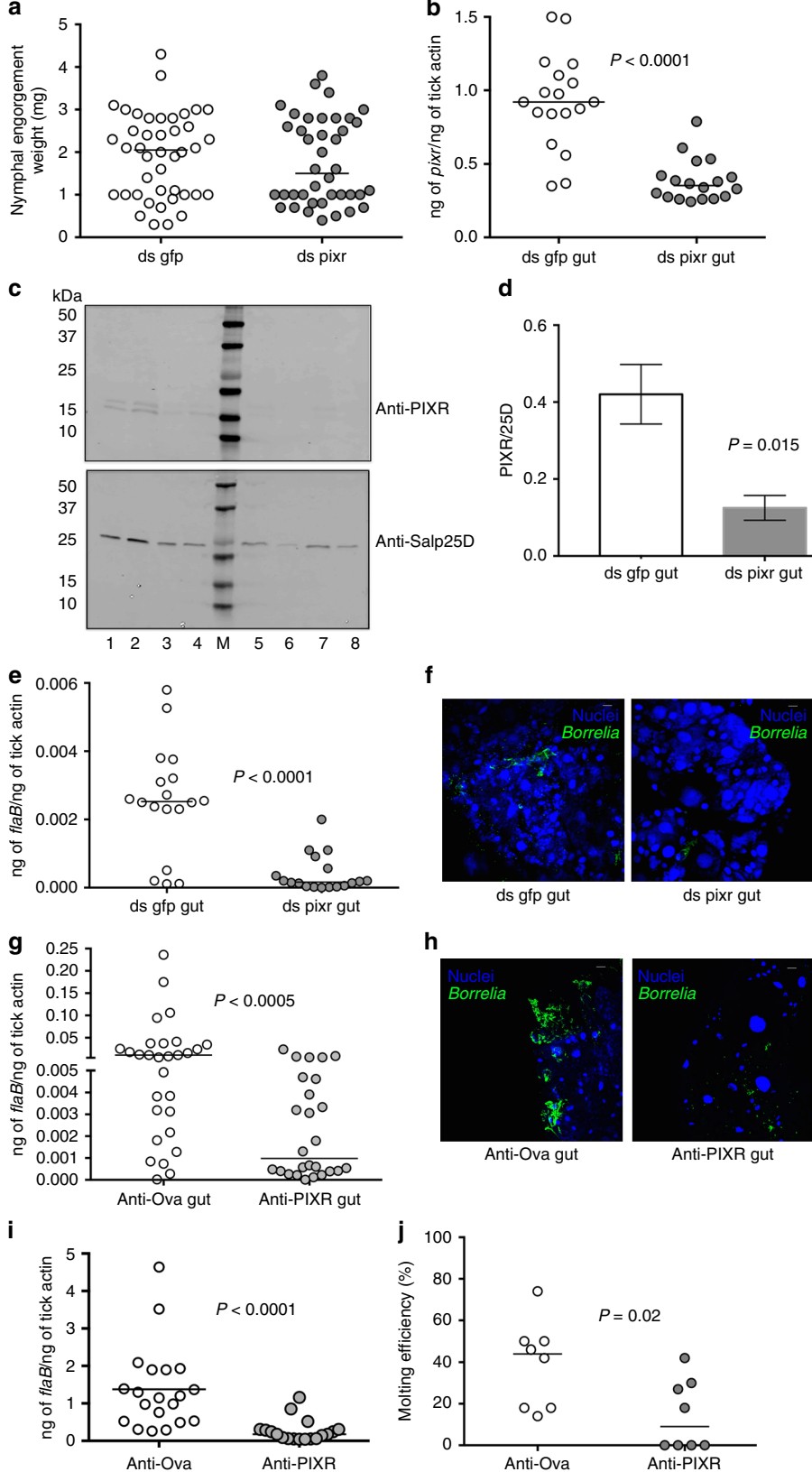

(Fig. 3a). qRT-PCR analysis also showed a significant decrease in *pixr* (Fig. 3b) as well as *pixr2* (Supplementary Fig. 1F) expression in the guts of ds *pixr* RNA-injected ticks when compared to that in control dsRNA-injected tick guts. Further, western analysis of

nymphal guts showed decreased expression of the PIXR/PIXR2 doublet in ds *pixr* RNA-injected ticks compared to control/ds gfp RNA-injected ticks (Fig. 3c and Supplementary Fig. 8C). Contrary to our expectation *pixr*-silenced nymphs showed decreased

*Borrelia* burden when compared to that in control ticks as seen by qRT-PCR (Fig. 3e) and by visualization of *B. burgdorferi* in nymphal guts by immunofluorescence microscopy (Fig. 3f). We also utilized an immunization approach to assess the role of PIXR in *B. burgdorferi* colonization of the nymphal gut. C3H/HeN mice were immunized with rPIXR or with Ovalbumin and infected with *B. burgdorferi* (N40), and immunized mice were challenged with nymphal ticks and ticks fed to repletion. Nymphal ticks that fed on PIXR-immunized *B. burgdorferi*-infected mice also showed decreased *Borrelia* burden when compared control ticks as seen by qRT-PCR (Fig. 3g) and by visualization of *B. burgdorferi* in nymphal guts by immunofluorescence microscopy (Fig. 3h), implying a potential function for PIXR in facilitating *B. burgdorferi* colonization of the tick.

**PIXR facilitates *B. burgdorferi* colonization and larval molting**. In its natural cycle, *B. burgdorferi* acquisition from the host and colonization of the gut occurs in larval stage[2]. RNAi approach for the larval stage is technically challenging, as due primarily to the small size, we observe increased mortality from trauma associated with microinjection. Therefore, we utilized an immunization approach to assess the role of PIXR in *B. burgdorferi* colonization of the larval gut as described for nymphal ticks. C3H/HeN mice immunized with rPIXR or with Ovalbumin (control) and infected with *B. burgdorferi* (N40) were challenged with larval ticks and ticks fed to repletion. qRT-PCR assessment of *Borrelia* loads in repleted larvae showed a marked reduction in *B. burgdorferi* burden in larvae that fed on PIXR-immunized mice compared to that in larvae fed on Ovalbumin-immunized mice (Fig. 3i), supporting that PIXR may have an important role during *B. burgdorferi* colonization of the tick. Further, larvae fed on rPIXR-immunized mice molted less efficiently to the nymphal stage (Fig. 3j) compared to larvae that fed on Ovalbumin-immunized mice.

**PIXR does not influence *B. burgdorferi* transmission to the murine host**. Transcript levels of *pixr* are increased in *B. burgdorferi*-infected nymphal gut (Fig. 2a); probably for this reason, RNAi-mediated knockdown of *pixr* transcripts in *B. burgdorferi*-infected nymphal guts did not efficiently decrease *pixr* transcript levels (data not shown). Therefore, to address the role of PIXR in *B. burgdorferi* transmission from the tick to the murine host we utilized the immunization approach. *Borrelia*-infected nymphs fed on PIXR-immunized or Ovalbumin-immunized mice, engorged comparably (Supplementary Fig. 2A). *B. burgdorferi* burdens in tick guts and salivary glands of ticks that fed on PIXR-immunized mice was also comparable to that in nymphs fed on Ovalbumin-immunized mice (Supplementary Fig. 2B). *B. burgdorferi* burdens in mouse skin at early (7 days and 10 days post-infection) and late (21 days post-infection) (Supplementary Fig. 2C) were also comparable between PIXR-immunized and Ovalbumin-immunized mice. These observations suggested that PIXR has a redundant role during *B. burgdorferi* transmission to the host.

**Bacterial viability is not affected by PIXR**. Tick gut microbiota have been shown to play a role in facilitating *B. burgdorferi* colonization[22]. Given the putative defense role of PIXR homologs in other arthropods, we postulated that PIXR might facilitate *B. burgdorferi* colonization of the tick gut by compromising the viability of members of the tick gut microbiota. Therefore, we examined the influence of rPIXR on the in vitro growth of *Staphylococcus aureus* and *Pseudomonas aeruginosa* as examples of gram-positive, and gram-negative bacteria respectively. PIXR had no significant impact on the growth of *S. aureus* or *P. aeruginosa* (Supplementary Fig. 3A, B) and was comparable to that in the presence of rIxophilin, a recombinant tick gut protein, also generated in the *Drosophila* expression system[23]. Further, rPIXR did not provide any direct growth advantage to *B. burgdorferi* as seen by comparable growth kinetics in the presence or absence of rPIXR (Supplementary Fig. 3C). Antimicrobial peptides defensin or cecropin were used as positive controls that demonstrated bactericidal activity against gram-positive (*S. aureus*) and negative bacteria (*P. aeruginosa*) respectively (Supplementary Fig. 3A, B). Gentamicin was cidal to *S. aureus*, and *P. aeruginosa* at 1–2 μg ml$^{-1}$ and to *B. burgdorferi* at 100 μg ml$^{-1}$ concentrations.

**PIXR affects bacterial biofilm architecture in the nymphal gut**. We proceeded to examine the tick gut by electron microscopy to determine if PIXR might influence the gross morphology of the tick gut. Since dissection of larval gut is hampered due to size, all microscopic visualization experiments were done using nymphal ticks. Nymphal ticks fed on ovalbumin or PIXR-immunized mice for 48 h were dissected and processed for transmission electron microscopy (TEM) or scanning electron microscopy (SEM). TEM visualization of the gut of clean nymphs fed on *B. burgdorferi*-infected mice, consistent with qRT-PCR assessment, showed decreased *B. burgdorferi* burden in the gut lumen of nymphs that fed on PIXR-immunized mice compared to that in the guts of nymphs that fed on Ovalbumin-immunized mice (Fig. 4b, c). The peritrophic matrix-like layer (PM) that straddles the gut epithelium and the gut lumen appeared dense in nymphs that fed on ovalbumin-immunized mice when compared to diffuse PM in the guts of nymphs fed on PIXR-immunized mice (Fig. 4b). Of note, ds *pixr* RNA-mediated knockdown of PIXR phenocopied the observations made on nymphs fed on PIXR-immunized mice (Fig. 4c).

SEM visualization of the guts of nymphal ticks fed on ovalbumin or PIXR-immunized mice showed the presence of bacterial biofilm-like structures in both groups of ticks. However, the morphology of the biofilm was dense and matte-like in the

---

**Fig. 3** PIXR facilitates *Borrelia* colonization and larval molting. **a** Nymphal engorgement weights in *pixr*-silenced and mock-injected nymphs. Each data point represents one tick: **b** qRT-PCR assessment of *pixr* transcript levels; **c** Western blot assessment of PIXR and Salp25D expression using a pool of two guts per lane. Lanes 1–4, ds *gfp* RNA-injected ticks and lanes 5–8, ds *pixr* RNA-injected ticks; protein markers (Precision Plus Biorad protein markers) are shown in lane M and their cognate molecular weights denoted to the left. **d** Quantification of PIXR expression normalized to that of Salp25D; **e** qRT-PCR assessment of *Borrelia flaB* levels in midguts; **f** visualization of *B. burgdorferi* burden by immunofluorescence microscopy using anti-*B. burgdorferi* antibody in the guts of ds *gfp* or ds *pixr* RNA microinjected pathogen-free nymphs after repletion on *B. burgdorferi*-infected mice; **g** *Borrelia flaB* levels in guts; and **h** Visualization of *B. burgdorferi* burden by immunofluorescence using anti-*B. burgdorferi* antibody in guts of nymphs repleted on ovalbumin-(anti-Ova) or rPIXR-immunized mice (anti-PIXR). Each data point in **b**, **e** and **g** represents a pool of two to three guts. **i** *Borrelia flaB* levels in larvae repleted on ovalbumin (anti-Ova) or rPIXR-immunized mice (anti-PIXR). Each data point represents a pool of five larvae. **j** Repleted larvae were allowed to molt to the nymphal stage and molting efficiency determined. Each data point represents the percentage of larvae that molted to nymphs. *Horizontal bars* represent the median. Mean values significantly different in a non-parametric Mann–Whitney test ($P < 0.05$) indicated. Scale bars, 10 μm (**f**, **h**)

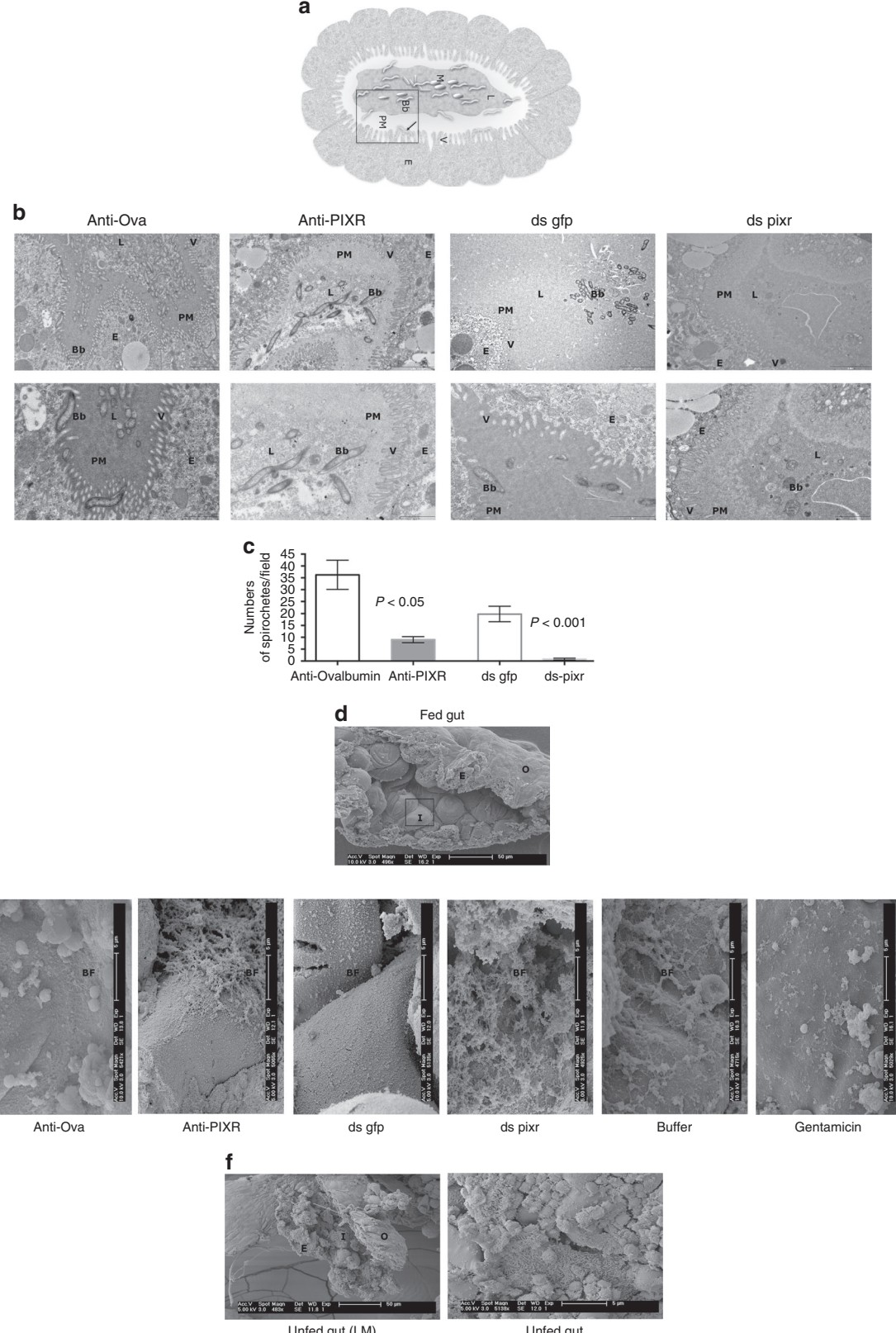

guts of nymphs that fed on PIXR-immunized mice when compared to sparse biofilms in the guts of nymphs that fed on Ovalbumin-immunized mice (Fig. 4e). ds *pixr* RNA-mediated knockdown of *pixr* also provided a similar increase in the biofilm-

like structures in nymphal guts when compared to that of control dsRNA-injected nymphs (Fig. 4e). SEM visualization of guts of nymphal ticks fed on PIXR-immunized and gentamicin-injected mice showed a decrease in the biofilm-like structures when

compared to that in PIXR-immunized and buffer-injected mice further, suggesting that these were bacterial biofilms (Fig. 4e). Finger-like projections decorated the lining of the unfed nymphal gut and biofilm-like structures were not apparent (Fig. 4f).

**PIXR inhibits gram-positive bacterial biofilm formation**. The SEM examination of nymphal guts suggested a role for PIXR in influencing tick gut bacterial biofilms. Therefore, we utilized an in vitro static biofilm assay to assess PIXR in the context of biofilm formation using *Staphylococcus aureus* as model biofilm-forming bacteria. While rPIXR did not affect the growth of *S. aureus* (Fig. 5a), rPIXR significantly decreased biofilm formation by *S. aureus* in a dose-dependent manner (Fig. 5b) when compared to rIxophilin[23]. However, rPIXR did not alter growth or biofilm formation by *Pseudomonas aeruginosa*, a potent biofilm-forming gram-negative bacterium (Fig. 5c, d) suggesting that PIXR preferentially inhibits biofilm formation among gram-positive bacteria.

To additionally determine if biofilms are increased upon PIXR abrogation, guts of nymphal ticks fed on ovalbumin or PIXR-immunized mice were processed for immunofluorescence (IFA) microscopy using antibodies against poly-N-acetyl glucosamine (PNAG). PNAG is an exopolysaccharide component of many gram-positive bacterial biofilms[24]. Indeed, increased reactivity to PNAG was readily observed in the guts of nymphs fed on PIXR-immunized mice compared to that in guts of nymphs fed on ovalbumin-immunized mice (Fig. 5e). *S. aureus*, a gram-positive bacterium, forms biofilms in which PNAG is a major component of the exopolysaccharide[25]. Therefore, we microinjected overnight planktonic grown *S. aureus* (~$4 \times 10^2$ bacteria/tick) into nymphal ticks and allowed them to feed for 48 h on PIXR-immunized or ovalbumin-immunized mice and guts dissected and processed for IFA using anti-PNAG antibodies. Reactivity to PNAG was observed in the guts of nymphs fed on PIXR-immunized mice compared to that in guts of nymphs fed on ovalbumin-immunized mice (Fig. 5f) suggesting that *S. aureus* biofilms were readily formed in the absence of PIXR in vivo.

**PIXR abrogation affects expression of gut immune response genes**. PIXR abrogation and the resultant increase in biofilms might lead to increased defense responses in the gut detrimental to *B. burgdorferi* survival and colonization. Expression profiles of known *I. scapularis* orthologues of arthropod defense response genes[9, 26, 27] were assessed by qRT-PCR. The expression levels of *stat*, encoding a transcription factor in the JAK-STAT pathway[27] that activates defense and repair response-genes[28], *relish* (Relish), encoding a key transcription factor of the IMD (immunodeficiency) pathway that responds to gram-negative bacteria[26], dual

oxidase (*duox*)[27], encoding DUOX that directly damages pathogens by the generation of reactive oxygen, and nitric oxide synthase (NOS)[29] encoding NOS that generates nitric oxide (NO) in response to septic insult or injury, were not altered (Fig. 6a–d). Transcript levels of *basket*, a key transcription factor of the JNK (Jun amino-terminal kinase) signaling pathway involved in development and in immune response to microbial challenge, stress and epithelial injury[30], were decreased upon PIXR abrogation, and transcript levels of *dorsal* a transcription factor of the Toll pathway[27] were significantly increased upon PIXR abrogation (Fig. 6e, f). The expression levels of antimicrobial peptides such as *scapularisin*5[18], as well as domesticated type VI secretion amidase effector[31] (*dae*), were reduced upon PIXR abrogation (Supplementary Fig. 4B, C). Transcript levels of four annotated *I. scapularis* peptidoglycan recognition proteins (PGRP) that bind bacterial peptidoglycan and activate immune pathways or neutralizes bacteria via catalytic amidase activity[27] were assessed and while transcript levels of two *pgrp*s that encode cytosolic proteins (ISCW004389 and ISCW024175 or *pgrp1* and *pgrp2*) were not altered upon PIXR abrogation, the expression levels of ISCW022212 or *pgrp4* that encodes a secreted PGRP was significantly decreased (Supplementary Fig. 4D–F). Transcripts for ISCW024689 that encodes a cytosolic PGRP were not detectable. Further, while the expression levels of *peritrophin*-1 and *peritrophin*-4, components of the PM-like layer in larval ticks[22, 32] were not altered, expressions of *peritrophin*-3 and *peritrophin*-5 were decreased in larval ticks that fed on PIXR-immunized mice when compared to that in ticks fed on ovalbumin-immunized mice (Supplementary Fig. 5C, E). *Peritrophin-2* transcripts were detected at very low and comparable levels in both groups (Supplementary Fig. 5B). Periodic acid Schiff's stain of nymphal tick guts to visualize the PM-like layer showed increased thickness in ticks that fed on PIXR-immunized mice compared to ticks that fed on control mice (Supplementary Fig. 5F–G) presumably due to increased biofilm formation and consequently increased extracellular polysaccharides.

**PIXR affects gut microbiome and metabolome composition**. We speculated that PIXR might influence the tick gut microbiome composition by regulating the ability of gram-positive bacteria to form biofilms in the gut when the tick takes a bloodmeal. We assessed the microbial composition of larval ticks by characterizing the 16S amplicons generated from ticks fed on PIXR-immunized or Ovalbumin-immunized mice using the Illumina sequencing method. The microbial composition of larval ticks that fed on PIXR-immunized mice showed differences at the phylum and at the genus level (Fig. 7a, b). The two groups of larvae indeed harbored significantly different bacterial microbiota as judged by weighted (analyses of similarity (ANOSIM) $R = 0.71$,

**Fig. 4** PIXR influences biofilm-like structures in tick guts. Guts of nymphal ticks fed for 48 h on ovalbumin-immunized (Ovalbumin) or rPIXR-immunized (PIXR) mice assessed by transmission electron microscopy (TEM): **a** pictorial depiction of the gut section; Lumen, L; Epithelium, E; *B. burgdorferi*, Bb; gut microbiota, M; and Villi, V. *Arrows* indicate *B. burgdorferi* that have breached the PM and are lying against the epithelial cells. A region similar to that represented in the *boxed area* is shown in the TEM image in **b**. Decreased numbers of *B. burgdorferi* in the gut lumen and a more diffuse peritrophic matrix (PM) in the rPIXR-immunized or ds pixr-injected guts compared to the dense PM in the Ovalbumin-immunized or ds gfp injected guts. Bottom panel in each column represents a focused view for clarity. Scale bars, 2 and 1 µm (top and bottom). **c** Quantification of the spirochete numbers in gut sections ($n = 6$ group). Scanning electron microscopy of 48 h-fed guts: **d** shown at low magnification (LM) of ~×500 is a representative gut slit open to reveal the inside of the gut (I), outside of the gut (O) and epithelium (E) lining the gut. Scale bar, 50 µm and regions similar to the *boxed area* is shown at a magnification ~×5000 in **e**. The inside of the gut showing sparse biofilms-like structures (BF) in ticks fed on ovalbumin-immunized mice (anti-Ova) or in ds gfp-injected guts (ds gfp) compared to the dense and matte-like biofilm in ticks fed on rPIXR-immunized mice (anti-PIXR) or ds *pixr* RNA-injected guts (ds pixr). Biofilm-like structures in PIXR-abrogated ticks fed on buffer-injected mice (buffer) are decreased in ticks fed on gentamicin-injected mice (Gentamicin); scale bar, 5 µm. **f** scanning electron microscopy of unfed guts: a representative gut slit open to reveal the inside of the gut (I), outside of the gut (O) and epithelium (E) (LM ~×480) and the inside of the gut showing finger-like projections at a magnification of ~×5000

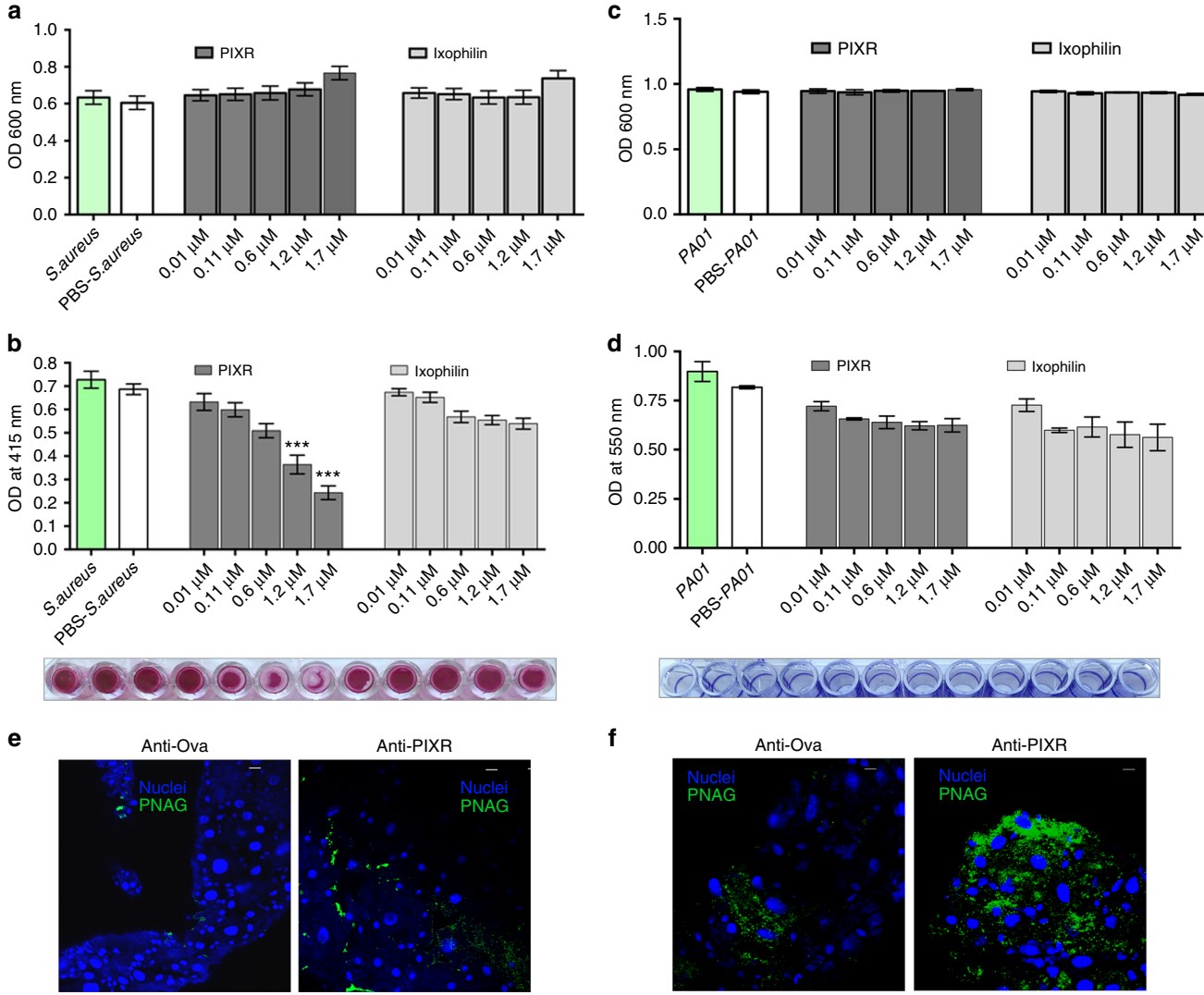

**Fig. 5** PIXR influences biofilms in vivo and in vitro. **a** Growth of *Staphylococcus aureus* cultures upon incubation with rPIXR or rIxophilin. **b** Biofilm formation of *S. aureus* in presence or absence of rPIXR or rIxophilin. Results are mean ± s.e.m. of three biological replicates. A representative image of the corresponding safranin-stained biofilm is shown below the graph. Bacteria cultured without any added recombinant protein or with PBS served as controls. Asterisks represent statistical significance determined by one-way analysis of variance. **c** Growth of *Pseudomonas aeruginosa* cultures upon incubation with rPIXR or rIxophilin. **d** Biofilm formation of *P. aeruginosa* in presence or absence of rPIXR or rIxophilin. Results are mean ± s.e.m. of three biological replicates. A representative image of the corresponding Crystal violet-stained biofilm (formed as a ring along the well-walls) is shown below the graph. Bacteria cultured without any added recombinant protein or with PBS served as controls. Visualization of poly-N-acetyl Glucosamine (PNAG) using rabbit anti-PNAG antibody: **e** in native biofilms; and **f** in biofilms formed upon injected *S. aureus* in guts of ticks fed for 48 h on ovalbumin-(anti-Ova) or rPIXR-immunized mice (anti-PIXR). Scale bars, 10 μm

$P = 0.001$) and unweighted (ANOSIM $R = 0.48$, $P = 0.001$) principal component assessment of beta diversity (Fig. 7c, d).

We assessed the composition of larval tick-associated microbiota as it is the stage most relevant to *B. burgdorferi* colonization in nature[2]. Most of the genera associated with the larval stage in this study have also been observed in the nymphal gut[33]. To ensure that viable bacteria were associated with the nymphal and larval ticks we performed qRT-PCR to assess the presence of specific bacterial genera in both nymphal and larval stage. Transcripts corresponding to the predominant bacterial genera observed by 16S rRNA amplicon sequencing of larval DNA were detected in both the larval and nymphal stage, albeit at differing levels in the two stages (Supplementary Fig. 7A–G). To additionally confirm that tick guts harbored viable bacteria we performed RNA-directed in situ hybridization using universal 16S rRNA probes to visualize bacteria in 48 h fed nymphal tick guts (control ds RNA-injected or fed on ovalbumin-immunized mice). We observed hybridization of the antisense probe to different regions of the diverticula (Fig. 7e). When PIXR was abrogated by anti-PIXR immunity or by *pixr* dsRNA we observed decreased hybridization signal (Fig. 7e), likely due to decreased accessibility of the probes upon increased biofilm formation further emphasizing the role of PIXR in altering bacterial architecture at the gut-lumen interface. Importantly, while levels of 16S rRNA transcripts corresponding to *Rickettsia* were not altered in nymphs or in larvae (Fig. 7f, h) levels of 16S rRNA transcripts of the genera *Brevibacterium* were significantly increased in the guts of nymphal ticks and in larval ticks (Fig. 7g, i) that fed on PIXR-immunized mice compared to that in ticks fed on ovalbumin-immunized mice. Interestingly, PIXR also decreased biofilm formation by *Brevibacterium caesi* in an in vitro biofilm inhibition assay without affecting the growth of *B. caesi* (Fig. 7j, k).

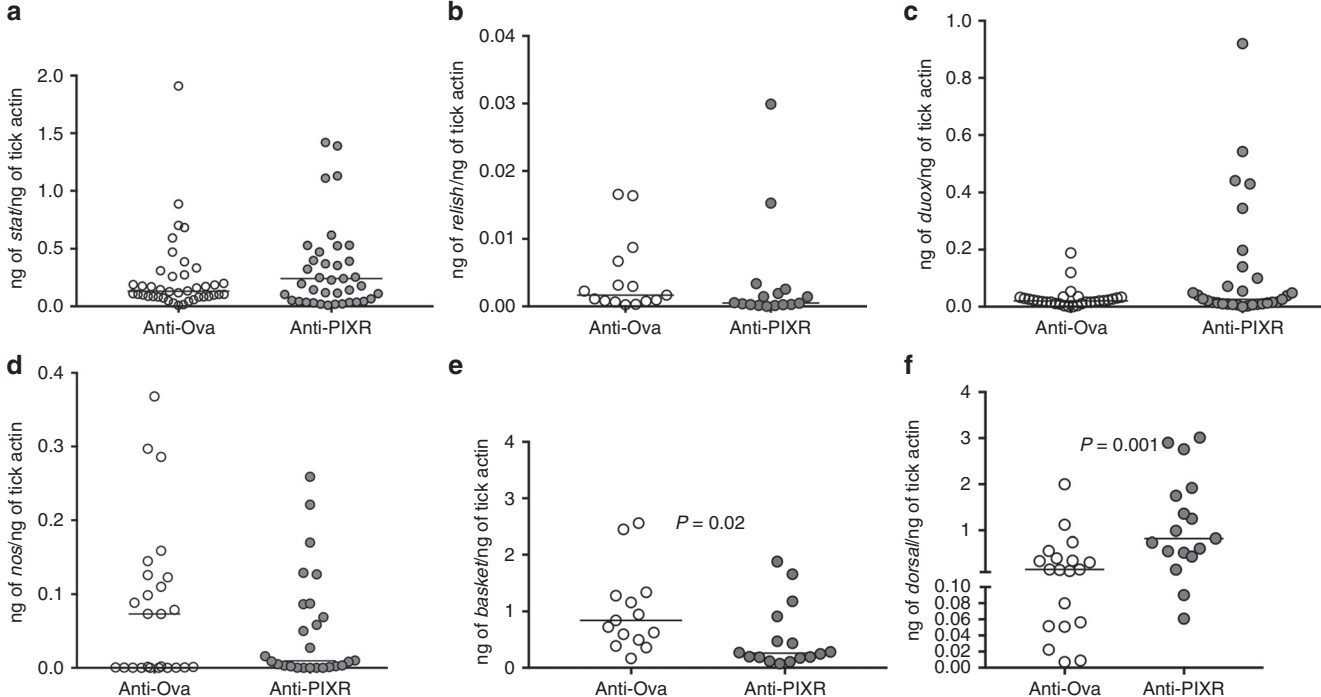

**Fig. 6** PIXR abrogation and impact on tick immune response genes. qRT-PCR analysis of expression levels of: **a** *stat*; **b** *relish*; **c** *duox*; **d** *nos*; **e** *basket*; and **f** *dorsal* in repleted larvae fed on rPIXR-immunized or ovalbumin-immunized *B. burgdorferi*-infected mice. Each data point represents a pool of two to three guts. Horizontal bars represent the median. Mean values significantly different in a non-parametric Mann–Whitney test ($P < 0.05$) indicated

Changes in the gut microbial members is likely to influence the metabolome of the tick gut due to differences in the metabolic functions unique to specific bacteria genera. We dissected guts from nymphal ticks fed on PIXR-immunized or Ovalbumin-immunized mice and processed gut supernatants for UPLC-MS-based identification of gut metabolites. A total of 59 metabolites were identified on Positive mode (73 hits in total), and 64 metabolites were identified on Negative mode (85 hits in total) and the integrated list of the 123 metabolites is shown in Supplementary Table 3. At least 38 and 39 differentially represented metabolites were identified under positive and negative mode, respectively, between the guts of ticks fed on PIXR-immunized or ovalbumin-immunized mice and is shown graphically by hierarchical clustering to heatmap (Fig. 8a). The identification of the differentially represented metabolites will require the use of additional strategies, which is beyond this article's scope.

**PIXR abrogation affects *B. burgdorferi* transcript levels.** As *B. burgdorferi* enters the tick from the mammalian host, the spirochete senses the gut environment by mechanisms that are not fully understood[34] and alters its transcriptome, turning on genes essential for colonization of the tick gut[2]. We speculate that gut metabolites might provide tick gut-specific cues for *B. burgdorferi* and impact the expression of spirochete genes required for colonization. A recent study by Iyer et al.[35] has delineated the global gene expression profiles of spirochetes in the various tick stages and in the mammalian host. We examined the expression profiles of a subset of spirochete genes shown to be specifically upregulated during spirochete entry from the murine host into the larval tick[35]. Genes *bb0690*, encoding a DNA binding protein shown to protect spirochetes from oxidative stress, *bba62*, encoding an outer membrane lipoprotein, and *bbb29* that encodes a glucose and maltose-specific phosphotransferase system[35, 36] were significantly altered (Fig. 8b–d) in larval ticks that fed on PIXR-immunized mice compared to ticks that fed on ovalbumin-

immunized mice. However, the expression levels of genes such as *bba15*, encoding for OspA, a lipoprotein critical for *B. burgdorferi* colonization of the tick gut[37], *bba74*, encoding an outer membrane associated protein expressed exclusively in the tick stage[38], and *rpoS*, that encodes for the alternative sigma factor RpoS-a key regulator of virulence gene expression[39], were comparable between larvae that fed on Ova-immunized or PIXR-immunized mice (Fig. 8e–g).

## Discussion

This study underscores the functional significance of the three-way interactions between the tick, its microbiome and the spirochete and offers new insights into *B. burgdorferi* colonization of the tick. We assessed the role of PIXR, a predicted secreted protein with a Reeler domain, that is induced upon feeding and upregulated in *B. burgdorferi*-infected tick guts, in the context of *B. burgdorferi* colonization and transmission. Homologs of PIXR are represented in several blood-feeding insects, and induced upon septic injury—hence suggested to be an immune response gene[19]. A family of at least 7 Reeler domain containing proteins are encoded in the *I. scapularis* genome of which only PIXR, PIXR2 and PIXR3 are predicted secreted proteins. These three paralogs are likely to have a functional consequence on *B. burgdorferi*, an extracellular pathogen. We could only detect PIXR and PIXR2 proteins in nymphal and larval stages. At the nucleotide level *pixr* and *pixr2* share 87% identity over a stretch of 548 bases (Supplementary Fig. 7) and ~70% identity at the protein level. Therefore, we could not selectively target the two paralogs by RNAi or by host immunity.

RNAi-mediated knockdown of *pixr* transcripts in nymphal ticks or abrogation of PIXR function by anti-PIXR immunity in larval and nymphal ticks resulted in decreased *B. burgdorferi* colonization. Building on TEM, SEM and IFA observations of nymphal guts and in vitro assays with rPIXR, we inferred that PIXR is an inhibitor of gram-positive bacterial biofilm formation,

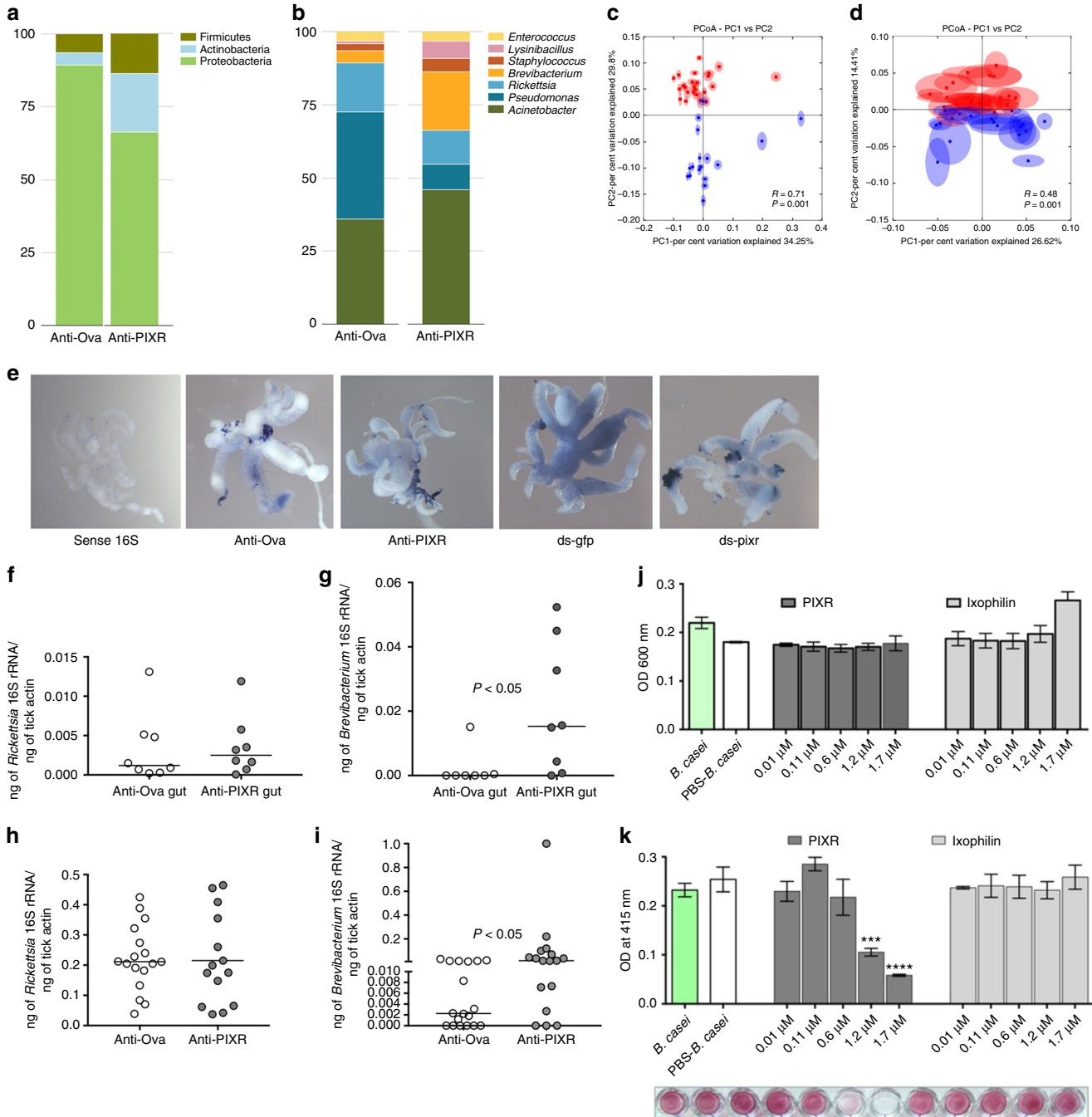

**Fig. 7** PIXR abrogation alters the composition of tick gut microbiota. 16S rRNA amplicon sequence analysis of microbial communities of larvae that fed on ovalbumin-immunized (control) or rPIXR-immunized mice (PIXR): **a** phylum; **b** genera; **c** principal coordinate analysis of weighted; and **d** unweighted jack-knifed UniFrac distances of microbial communities between PIXR (*blue*) and control (*red*) larval ticks. Statistical significance determined by ANOSIM is indicated within each PCA plot; **e** whole-mount in situ hybridization of universal 16S rRNA probes in 48 h fed guts of nymphs fed on PIXR-immunized (anti-PIXR) mice or ds pixr-injected (ds pixr) nymphs showed restricted staining compared to that in guts of nymphs fed Ovalbumin-immunized (anti-Ova) or in ds gfp-injected (ds gfp) nymphs. qRT-PCR assessment of 16S rRNA transcript levels of: **f** *Rickettsia*; **g** *Brevibacterium* in fed nymphal guts; **h** *Rickettsia*; and **i** *Brevibacterium* in repleted larvae. Each data point in **f**–**i** represents a pool of three nymphal guts and five larval ticks respectively. Mean values significantly different in a non-parametric Mann–Whitney test (*P* < 0.05) indicated. **j** Growth of *Brevibacterium casei* cultures upon incubation with rPIXR or rIxophilin. **k** Biofilm formation of *B. casei* in presence or absence of rPIXR or rIxophilin. Results are mean ± s.e.m. of three biological replicates. A representative image of the safranin-stained biofilm is shown below the graph. Bacteria cultured without any added recombinant protein or with PBS served as controls. Asterisks represent statistical significance by one-way analysis of variance

and that PIXR possibly functions in vivo by limiting the expansion of gram-positive bacterial biofilms in the tick gut. *Ixodes* anti-freeze glycoprotein or IAFGP, another inhibitor of gram-positive bacterial biofilms that is specifically induced by

*Anaplasma phagocytophilum* infection, was shown to inhibit biofilm formation by binding to cell wall peptidoglycan[33] that is exposed in gram-positive bacteria. How PIXR selectively inhibits gram-positive bacterial biofilms remains to be examined.

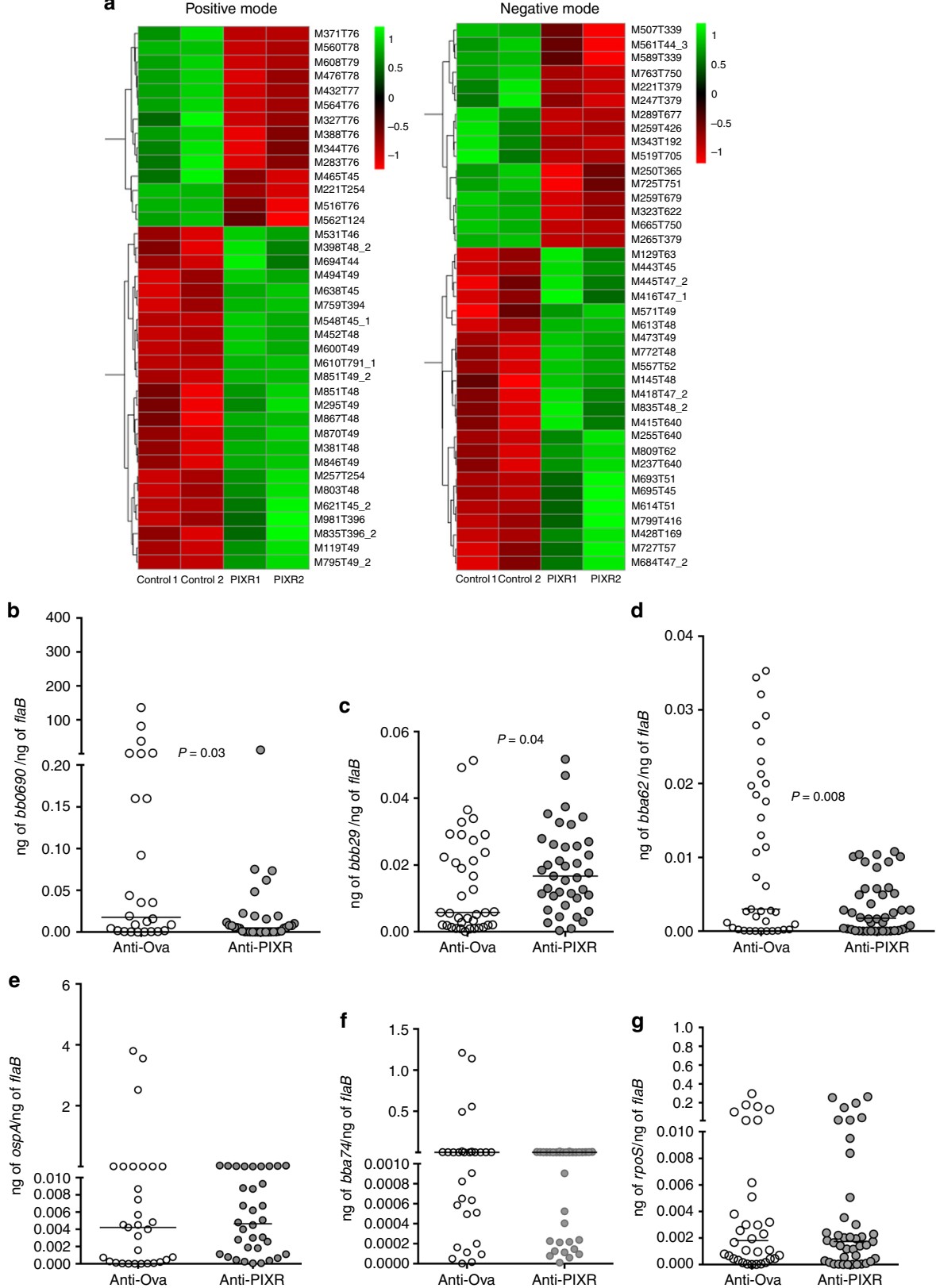

Larval ticks that fed on PIXR-immunized mice demonstrated differences in the microbiome composition when compared to larvae that fed on control (ovalbumin-immunized) mice. In the absence of PIXR, gram-positive bacterial biofilms likely increase and consequently shift the microbiome composition. Seven predominant genera were observed in larval ticks. In our previous study[22], we observed additional genera including *Delftia*, *Commamonas*, *Stenotrophomonas* and *Acidovorax*. Bacterial genera such as *Commomonas*, *Acidovorax* and *Stenotrophomonas* have been suggested to be common environmental bacteria[40]. The

current study has utilized additional criteria as suggested by Salter et al.[40] to exclude potential nonspecific amplicons. Batch variations between the ticks used in this study and the previous study, differences in DNA extraction protocols, sequencing platform and analysis parameters, could also account for some of the differences between the current and the earlier study[22]. Nonetheless, we note that different studies observe varying diversities in the microbiota compositions of tick species with some showing as few as 2–10 bacterial genera[33, 41–48] and some showing greater than 20 bacterial genera[49–53]. Several factors including tick species, gender, geographic location[44], developmental status, time since molting and feeding[54] are expected to influence the composition of the microbiome. Using genus-specific 16S rRNA primers to assess viable bacteria by qRT-PCR we observed that bacteria such as *Pseudomonas*, *Acinetobacter* and *Brevibacterium* found commonly in the environment[40] are associated with larval and nymphal ticks and whole-mount in situ hybridization also suggested the presence of these bacteria in nymphal guts (Supplementary Fig. 6). Although endosymbionts such as *Rickettsia*[42, 48] might represent predominant and stably associated bacteria, we cannot rule out the possibility that tick guts might also engage with environmental bacteria[43] and with host-skin associated bacteria during blood-feeding and these associations may have functional consequences on pathogen transmission and colonization. Progress in the functional characterization and visualization of tick-gut-associated bacteria will facilitate a better understanding of bona-fide members of the tick microbiome. Until then, we refer to these bacteria as tick-associated microbiota.

We show that changes in the composition of tick-associated microbiota as a consequence of PIXR abrogation might impact *B. burgdorferi* colonization in multiple ways. Increased biofilms, and alterations in the microbiota composition in conjunction with a diffuse PM barrier might provoke immune responses detrimental to *B. burgdorferi*. An increase in the transcript levels of *dorsal*, a transcription factor that activates effector genes upon Toll-mediated signaling via gram-positive bacteria[27] is consistent with increased gram-positive bacteria such as *Brevibacterium* and *Lysinibacillus* upon PIXR abrogation. Increased expression of *dorsal* might upregulate novel immune responses detrimental to *B. burgdorferi*. PIXR abrogation also shows a significant reduction in the expression levels of a subset of immune effector genes. Excreted polysaccharides of biofilms have been shown to have immune suppressive effects in the mammalian host[55, 56]. Biofilm increase upon PIXR abrogation might potentially suppress some immune responses of the tick. The JNK pathway regulates critical physiological processes other than defense responses[30] and the downregulation of *basket*, a key transcription factor in this pathway might affect aspects of tick physiology relevant to *B. burgdorferi* colonization. It is also possible that increased biofilms present a physical barrier to *B. burgdorferi* colonization.

Spirochetes in larvae that fed on PIXR-immunized mice show decreased levels of a subset of genes shown by Iyer et al.[35] to be preferentially increased during spirochete entry into the larval tick. Cyclic-diguanylate monophospate (c-di-GMP), a second messenger molecule, and two two-component histidine kinase sensor systems (HK1-RRP1 and HK2-RRP2) have been implicated in transducing the environmental signals and triggering changes in gene expression critical for spirochete survival in the diverse milieus[34]. PIXR abrogation leads to changes in the composition of gut metabolites and raises the possibility that gut/microbial metabolites might influence *B. burgdorferi* gene expression. There is increasing evidence that NO regulates c-di-GMP levels and regulates biofilm formation[57]. Given the role of c-di-GMP in *B. burgdorferi* gene expression[34], it might be relevant to assess by targeted metabolomics assays if increased gram-positive bacterial biofilms might impact NO levels in the gut and impact *B. burgdorferi* colonization.

How this laboratory finding relates to the microbiota associated with larval ticks and infection prevalence in endemic areas is critical. Although, field-collected ticks have been shown to predominantly harbor *Rickettsia*[42] several bacterial genera including *Brevibacterium* observed in this study have also been observed in field-collected ticks, predominantly nymphal and adult ticks[44, 51, 58–60]. To date, the presence or absence of specific genera have not been associated with *B. burgdorferi* prevalence. While specific genera may vary between studies depending on the environment, common functional elements of these bacteria may provide similar impacts on the tick and the pathogen/s it harbors.

PIXR also plays a role in larval molting. Tick gut-associated microbiota appear to have a propensity to form biofilms during feeding as we could not see biofilm-like structures in unfed tick guts. Increased biofilms, especially, by gram-positive bacteria, might be a physical nuisance for the tick confounding blood digestion, and nutrient availability essential for development and molting. The tick gut thus has a vested interest in managing its gut microbiota. Importantly, the pathogens that *I. scapularis* encounters traffic through the gut and exploit the molecular interactions between the tick gut and the microbiota to colonize the tick. The tick gut-associated microbiota might present a 'barricade' or a 'gateway' to incoming pathogens depending on the functional genome of the bacterial members that predominate. How the *I. scapularis* gut manages its microbiota might therefore be a critical prelude that determines pathogen survival in the tick. Understanding the molecular biology of gut microbial homeostasis is likely to uncover new paradigms in tick-pathogen interactions.

## Methods

**Ticks and animals.** *I. scapularis* nymphs and larvae were obtained from a tick colony at the Connecticut Agricultural Experiment Station in New Haven, CT, USA. Ticks were maintained at 23 °C and 85% relative humidity under a 14 h light, 10 h dark photoperiod. *Borrelia*-infected nymphs were generated by placing larvae on *B. burgdorferi*-infected C3H mice, and fed larvae were molted to nymphs. The *B. burgdorferi* strain used was the N40 strain (Passage 12), originally isolated from a tick collected in New York[61]. For active immunization studies, 4–6 weeks of age, female C3H/HeJ mice (Jackson Laboratory) were used. The protocol for the use of mice was reviewed and approved by the Yale Animal Care and Use Committee.

**Subtractive hybridization.** At least 15–20 *Borrelia*-infected unfed *I. scapularis* nymphs were individually tested by DNA-PCR for *Borrelia* infection using the *flab* primers that amplify the *flagellin* gene essentially as described earlier[22] and a batch

**Fig. 8** PIXR influences the tick gut metabolome and impacts *B. burgdorferi* gene expression. **a** Heat map showing alterations in the profile of differentially represented gut metabolites between ticks fed on PIXR-immunized (PIXR 1 and 2-each sample representing a pool of 10 tick guts) or ovalbumin-immunized mice (controls 1 and 2 each-sample representing a pool of 10 tick guts) in the positive and negative mode. Normalized signal intensities are shown as a color spectrum, green indicating low abundance and red indicating high abundance (least to abundant from −1 to +1); qRT-PCR assessment of the expression levels of *B. burgdorferi* genes: **b** *bb0690*; **c** *bbb29*; **d** *bba62*; **e** *ospA*; **f** *bba74*; and **g** *rpoS* in larvae fed on *B. burgdorferi*-infected ovalbumin-immunized (control) or rPIXR-immunized mice (PIXR). Horizontal bars represent medians. Mean values significantly different in a non-parametric Mann–Whitney test ($P < 0.05$) indicated

of nymphs with an infection rate of 90–95% was chosen as the tester group for subtractive hybridization. An age-matched (nymphs molted around the same time as the *Borrelia*-infected nymphs) batch of clean nymphs was chosen as the driver group for subtractive hybridization. At least 30–50 nymphs from each group were allowed to feed on pathogen-free C3H/HeN mice for 72 h. Subsequently, nymphs were collected, midguts were dissected and processed for total RNA extraction as described earlier[23]. cDNA was generated from 2 μg of RNA from clean and *Borrelia*-infected midguts using the Super SMART™ PCR cDNA synthesis kit according to the manufacturer's instructions (Clontech, CA). The cDNA was then directly utilized in the forward subtraction procedure using the PCR-Select cDNA Subtraction kit manual (Clontech). cDNA prepared from the *Borrelia*-infected guts served as the 'Tester' and cDNA prepared from the clean guts served as the 'Driver' in the subtraction procedure to enrich for cDNAs preferentially expressed and upregulated in the *Borrelia*-infected tick gut. The cDNAs were then cloned into a T/A cloning vector and plated on LB-Ampicillin plates. At least 500 single colonies were picked and dot-blot nitrocellulose membrane arrays of the clones generated and screened using labeled probes of the Tester (*Borrelia*-infected gut cDNA) and Driver (clean gut cDNA) cDNAs as described in the PCR-Select Differential Screening kit (Clontech). About 200 clones that hybridized preferentially with the Tester and not the Driver probes were considered differentially expressed in the *Borrelia*-infected guts. Positive clones were then picked, plasmid purified and insert DNA sequenced at the W.M Keck Sequencing Facility at Yale University using the nested PCR primers flanking the inserts. In silico analysis of the DNA sequences was performed using the Basic Local Alignment Search Tool (www.ncbi.nlm.nih.gov/blast) to search for homology with genes described in the public database.

**Tick RNA isolation and quantitative RT-PCR**. Ticks were allowed to feed to repletion. Nymphs were dissected and salivary glands and midguts were pooled (three ticks), homogenized and RNA was extracted using the Trizol RNA extraction procedure (Invitrogen, CA). The same procedure was performed to assess guts of unfed ticks. cDNA was synthesized using the iScript RT-PCR kit (Biorad, CA) and analyzed by qPCR for the expression of *pixr*, and *pixr2, 3, 4, 5, 6*, and 7, using the iQ Sybr Green Supermix (Biorad) on a MJ cycler (MJ Research, CA) and primers shown in Supplementary Table 2. Live *B. burgdorferi* burdens were assessed in the cDNA samples using *B. burgdorferi*-specific *flaB* primers described earlier[12]. Tick immune response genes, *stat, relish, dorsal, nos, duox, basket, scapularisin1* and 5, *dae, pgrp1, 2, 3 and 4*, were assessed by qRT-PCR using primers listed in Supplemental Table 2 and of *peritrophin1, 2, 3, 4* and 5, components of the peritrophic matrix-like layer of the tick gut as described earlier[22]. A subset of *B. burgdorferi* genes (*bb0690, bbb29, bba62, bba74, bba16 (ospA)* and *rpoS* upregulated during spirochete entry into the tick from the murine host was assessed by qRT-PCR using primers described by Iyer et al.[35] and listed in Supplementary Table 2.

**Protein expression analysis by immunoblotting**. To assess tissue-specific expression equal amounts (~2 μg) of 66–72 h-fed nymphal midgut extract (gut), and SGE were each electrophoresed on a sodium dodecyl sulphate (SDS) 12% polyacrylamide gel and transferred to nitrocellulose membranes. The membranes were blocked with phosphate-buffered saline (PBS) containing 5% milk powder and the immunoblots were probed with a 1:500 dilution of polyclonal rabbit anti-rPIXR serum. Polyclonal rabbit anti Salp25D, a protein expressed in the gut and salivary glands[62] or Salp14, a predominantly salivary gland-specific protein[63] were used to assess tissue-specific expression. Immunoreactive bands were visualized using Infrared (IR) Dye800CW-conjugated goat anti-rabbit secondary antibodies (LI-COR, NE) and the LI-COR 9100 imaging system. To assess and compare PIXR expression in larval and nymphal guts in presence and absence of *B. burgdorferi*, equal amounts (~2 μg) of protein extracted from fed nymphal midgut (gut), pools of 5–6 fed larval ticks, pools of 2–3 unfed clean or *B. burgdorferi*-infected nymphal guts, and pools of ~30–40 unfed larval ticks were each electrophoresed on a SDS 12% polyacrylamide gel and transferred to nitrocellulose membranes and probed with anti-PIXR or anti-Salp25D antibodies and bound antibodies detected as described above. Salp25D expression was not significantly altered upon infection with *B. burgdorferi*[62, 64] and served as a control. Protein expression of PIXR, as seen by western blot analysis, was quantified using ImageJ 1.50 (imageJ.nih.gov/ij) and normalized to the expression of Salp25D.

To identify the PIXR paralogs in the ~15 kDa doublet fed nymphal gut or larval protein extracts in PBS were electrophoresed on a 7–20% gradient SDS-polyacrylamide gel, gel stained with Coomassie blue and the doublet band cut out with a clean razor blade. The proteins in the gel bands were in-gel tryptic digested and processed for LC-MS/MS at the Yale Proteomic Facility using a Waters/Micromass AB QSTAR Elite spectrometer. All MS/MS spectra were searched in-house using the MASCOT algorithm and the MASCOT Distiller program (http://www.matrixscience.com) and peak list searched against the *I. scapularis* genome database.

**Cloning of the full-length *pixr2***. To confirm the full-length sequence of *pixr2* we utilized the forward primer specific to the 5'-end of ISCW01773 including the ATG start site of the gene (pixr2FL-forward) and a reverse primer containing the 3'-end of the gene including a five-base poly A tag (Supplementary Table 1). This also conforms to the 3'-end of the sequence of the expressed sequence tag

G894P5138RP5.T0 that shares 97% identity with ISCW017733. These primer sets were used to specifically amplify transcripts corresponding to ISCW01772 from cDNA templates prepared using total RNA isolated from fed larval or nymphal guts. A specific amplicon of ~550 base pairs were obtained from both larval and nymphal gut RNA. The amplicons were cloned into pGEM Teasy vector (Promega, WI) and at least four recombinant plasmid clones representing transcripts of larval or nymphal gut sequenced at the W.M Keck Facility using the Sanger sequencing method. The obtained sequences were identical and were then compared against the public database using the BLASTN suite (National Centre for Biotechnology Information, NCBI). The sequence was translated using the Translate tool (web.expasy.org) and the DNA and protein sequence aligned with PIXR and visualized using the T-Coffee multiple alignment suite (tcoffee.crg.cat or http://www.ebi.ac.uk/Tools/msa/tcoffee/).

**Production of recombinant PIXR and generation of antibodies**. The mature protein coding region of *pixr* was amplified using fed tick gut cDNA as the template, cloned into the pMT/Bip/V5-HisA plasmid (Invitrogen), and validated by sequencing. *Drosophila melanogaster* S2 cells were transfected with the plasmids containing PIXR and stable transfectants were generated. Protein expression was induced with copper sulfate as described by the manufacturer (Invitrogen). The supernatant was filtered using a 0.22 μm filter (Millipore, MA) and rPIXR were purified from the supernatant using Ni-NTA Superflow column chromatography (Qiagen, CA) and eluted with 250 mM imidazole. The eluted fractions were sterilized using a 0.22 μm filter, concentrated with a 5 kDa concentrator (Sigma-Aldrich, MO) and dialyzed against PBS. The purity of rPIXR was assessed by Coomassie blue staining and protein concentration determined using BCA protein assay kit (Thermo Fisher Scientific Inc., IL). To detect glycosylation on rPIXR the Glycoprotein Detection kit GLYCOPRO was used as per the instructions of the manufacturer (Sigma-Aldrich, MO). Horseradish peroxidase (HRP) (0.5 μg per lane) (Thermo Fisher scientific) was used as a positive control[65] and recombinant Glycerophosphodiester phosphodiesterase (1 μg per lane) (r GlpQ)[66], a bacterial protein, was used as a negative control for presence of glycosylation.

To generate polyclonal antibodies against PIXR, two New Zealand white rabbits were immunized in complete Freund's adjuvant with 30 μg of rPIXR followed by two booster immunizations with 30 μg of rPIXR once every 3 weeks in incomplete Freund's adjuvant. The rabbits were euthanized 3 weeks after the second booster immunization and blood collected for serum. Pre-bleeds and test bleeds were obtained at or before each immunization schedule for assessing antibody titers to rPIXR by ELISA or immunoblot assay.

**RNAi silencing of *pixr***. Primers were designed by addition of a T7 promoter site (5'-TAATACGACTCACTATAGGGAGA-3') at the 5'-end of the forward (5'-GCTACCGTCTCGTCCAAAAC-3') and reverse (5'-TTCTTCGAGGTGTGCGT-GAT-3') primers. dsRNA complementary to *pixr* was synthesized by using the MEGAscript RNAi kit and dsRNA was purified and quantified spectroscopically (Ambion). dsRNA (0.5 μl) was injected into the anal pore of *B. burgdorferi*-infected or pathogen-free nymphs using 10 μl microdispensers (Drummond Scientific, Broomall, PA) as described earlier[62]. Control nymphs were injected with an irrelevant dsRNA complementary to green fluorescent protein (gfp) encoding gene or with dsRNA elution buffer as described earlier[22].

To assess colonization of *B. burgdorferi*, mice were subcutaneously injected with 100 μl of *B. burgdorferi* (N40) spirochetes ($10^5$ ml$^{-1}$). 3 weeks post-injection ear skin biopsies were obtained from each animal and *B. burgdorferi* burden assessed by QPCR as described earlier[22]. Comparably infected mice (three mice per group) were challenged with ~15–20 pathogen-free *dspixr* or *dsgfp-injected*. *I. scapularis* nymphs and ticks allowed to feed to repletion. Total RNA was isolated from pools of 3 guts and transcript levels of PIXR and *B. burgdorferi* burden assessed by QRT-PCR as described earlier[22] and transcripts levels were normalized to tick actin.

**Active immunization with recombinant PIXR**. C3H/HeN mice were immunized subcutaneously with 3 doses containing 5 μg of rPIXR emulsified with Complete Freund's Adjuvant (first dose) and two subsequent booster injections emulsified in Incomplete Freund's Adjuvant at 3-week intervals. Control mice were inoculated with adjuvant and Ovalbumin (5 μg). After completion of the immunization regimen, test bleeds were obtained to confirm seroreactivity to rPIXR or Ovalbumin by western blot as described above. Mice were then subcutaneously injected with 100 μl of *B. burgdorferi* (N40) spirochetes ($10^5$ ml$^{-1}$). Three weeks post-injection ear skin biopsies were obtained from each animal and *B. burgdorferi* burden assessed by qPCR as described earlier[22]. Comparably infected rPIXR-immunized or Ova-immunized mice were challenged with approximately 200 pathogen-free *I. scapularis* larvae and larvae allowed to feed to repletion. Total RNA was isolated from pools of 5 larvae and *B. burgdorferi* burden assessed by qRT-PCR as described earlier[22].

To assess transmission, at least 4 *B. burgdorferi*-infected nymphs were placed on each PIXR-immunized or Ova-immunized mouse and 5 mice used in each group and ticks were allowed to feed to engorgement, weighed and guts and salivary glands (SG) dissected for mRNA isolation and *B. burgdorferi flaB* transcripts assessed by qRT-PCR as described above. *B. burgdorferi* burden in mice 5, 10 and 21 days post engorgement was also assessed as described[12].

**In vitro assessment of the effect of rPIXR on bacterial viability.** To test growth inhibitory effects of rPIXR on *Staphylococcus aureus* strain SA113 (ATCC 35556), derived from laboratory strain NCTC 8325, *Pseudomonas aeruginosa* strain PA01 and *B. burgdorferi* N40 (Passage 12), bacteria were grown to log phase and diluted to approximately $10^6$ and $10^5$ bacteria per ml respectively in glucose-supplemented tryptic soy broth for *S. aureus*, Luria Bertani broth plus 1% glucose for *P. aeruginosa* or BSK-H medium for *B. burgdorferi* N40. Bacterial suspensions were incubated with rPIXR or rIxophilin, a secreted tick gut protein generated in the *Drosophila* expression system and shown to be an inhibitor of thrombin activation[23] as a control, in PBS at a final concentration of 2 µM or human Defensin HNP2 (Sigma-Aldrich) in PBS at 25 µg ml⁻¹ or Cecropin A (Sigma-Aldrich) in PBS at 10 µg ml⁻¹ or Gentamicin (Life Technologies) at 2 µg ml⁻¹ for *P. aeruginosa* and *S. aureus* and 100 µg ml⁻¹ for *B. burgdorferi*. *P. aeruginosa* and *S. aureus* were incubated at 37 °C and *B. burgdorferi* at 30 °C and growth was assessed at several time points. Aliquots of 50 µL were withdrawn at 4, 8 and 24 h and optical density as a measure of growth measured at 600 nm for *P. aeruginosa* and *S. aureus*. Assuming, OD of 1.0 = $10^8$ CFU ml⁻¹ the data was converted from Absorbance at 600 nm into CFU per ml and analyzed. For *B. burgdorferi*, aliquots of 50 µl were withdrawn at 24, 48 and 72 and 96 h, diluted 10-fold and spirochetes counted using a Petroff Hausser counter under a dark-field microscope.

**In vitro assessment of the effect of rPIXR on bacterial biofilms.** Biofilm assays were performed as described[67]. Briefly, planktonic overnight cultures of *Staphylococcus aureus* strain SA113 (ATCC 35556), derived from laboratory strain NCTC 8325, *Pseudomonas aeruginosa* strain PA01 or *Brevibacterium casei* (ATCC 35513) were diluted in glucose-supplemented tryptic soy broth for *S. aureus*, Luria Bertani broth plus 1% glucose for *P. aeruginosa* or nutrient broth (1% glucose) for *B. casei* to OD$_{600}$ of 0.015 (approximately $1.5 \times 10^6$ cells per ml) and plated into 96 well plates (Corning) in the presence of rPIXR (0.01–1.7 µM). Control wells received rIxophilin[23]. Plates were incubated without shaking for 18 h at 37 °C (for *S. aureus* and *P. aeruginosa*) or 30 °C (for *B. casei*). Thereafter, bacterial growth was assessed at OD$_{600}$ (BioTek ELISA plate reader). The supernatants were then aspirated, and the wells washed twice with water. Bacterial biofilms adhered to the bottom of the wells and were dried and stained with safranin for *S. aureus* or *B. casei* or with crystal violet for *P. aeruginosa*. The dye was then dissolved in 33% acetic acid and absorbance quantified at 415 nm (BioTek ELISA plate reader).

**Electron microscopic assessment of the effect of PIXR on gut biofilms.** *I. scapularis* nymphs were allowed to feed on rPIXR or Ovalbumin-immunized *B. burgdorferi*-infected mice for 48 h. Repleted ticks could not be processed due to interference from excessive bloodmeal. The ticks were then carefully detached from the mice and dissected in 2% PFA and 2.5% Glutaraldehyde. The guts were gently rinsed and placed in 2% PFA and 2.5% Glutaraldehyde containing 0.05% Ruthidium red for half an hour at room temperature and half an hour at 4 °C, rinsed in PBS, dehydrated in an ethanol series, embedded in epoxy resin, hardened and ultra-thin sectioned for transmission electron microscopic visualization (TEM) on a FEI Tencai Biotwin transmission electron microscope at 80 Kv. Images were taken using Morada CCD and iTEM (Olympus) software. For SEM ticks were similarly allowed to feed on mice and guts dissected. The guts were slit open longitudinally with a fine razor blade so as to expose the luminal side of the gut and processed as for TEM but after dehydrating, guts were dried in Leica 300 critical point dryer, glued to aluminum stubs and sputter coated with 15 nm of gold before visualization in a FEI ESEM using 5–10 kV at a working distance of 12 mm.

Comparably infected mice (three mice/group) were challenged with approximately 10 pathogen-free *dspixr* or *dsgfp*-injected. *I. scapularis* nymphs and ticks allowed to feed for 48 h. The ticks were then carefully detached from the mice and dissected in 2% PFA and 2.5% Glutaraldehyde and processed for electron microscopy as described above. For counting spirochetes, at least 5–6 individual ticks were examined /group and 2 fields examined/section.

**Sectioning of nymphs for Periodic acid Schiff's staining.** Nymhal ticks were fed on Ovalbumin or PIXR-immunized mice for 72 h, fixed in Carnoy's fixative for 1 h, washed in graded alcohol followed by 3 washes in Xylene, paraffin embedded and sectioned at approx. 5 µm tickness. Sections were stained with Periodic acid Schiffs base (PAS) at the Yale Histology Core facility as described earlier[22] and visualized under a Zeiss Axio YLCW023212 microscope at 40X magnification using the ZEN Lite software (Carl Zeiss Inc, NY). The thickness of the PAS positive PM-like layer was assessed in 3 different regions in each microscopic field, and arithmetic average computed. At least 8–10 individual ticks were examined /group and 3 fields examined/section.

**Assessment of the microbial composition of the tick gut.** Larval ticks were fed to repletion on Ovalbumin-immunized or PIXR-immunized mice, surface sterilized in 70% ethanol for 10 min and washed three time in distilled water prior to DNA extraction. Larvae were pooled in groups of 5 and DNA extracted using the PowerSoil DNA isolation kit (MoBio Labs, CA). Buffer used in the extraction was also processed to determine bacteria present in the reagents and supplies used in amplicon generation and processing. Bacterial 16S rRNA amplicons were then generated using barcoded 16S universal primers (515F/806R) and protocols

outlined by the Earth Microbiome Project (www.earthmicrobiome.org). The amplicons were sequenced on an Illumina MiSeq system that generated 251-base paired-end reads and the sequences were quality filtered, paired-end joined and de-multiplexed using QIIME version 1.8.0 [www.qiime.org]. These reads were clustered and operational taxonomic units (OTUs) were picked using a two-step pipeline[68] where an initial 'closed-reference' OTU picking was done by mapping against the Greengenes database (gg_13_8_otus.tar.gz from ftp://greengenes.microbio.me) and then the remaining reads were clustered de novo before picking the OTUs against the same Greengenes database. An average of 35,000 reads and ~8000 reads were obtained in each of the tick samples and in the buffer sample respectively.

To carry out the phylogenetic analysis of the sequences we first generated a phylogenetic tree from a set of representative sequences for each OTU in the study, using multiple sequence alignment in FastTree[69]. Using this tree and the OTU table that contained the number of sequences in each OTU and each sample, both quantitative (weighted UniFrac) and qualitative (unweighted UniFrac) distance metrics were computed[70] and used in the beta diversity analysis that characterize the microbial communities represented in larvae that fed on PIXR-immunized mice compared to larvae that fed on Ovalbumin-immunized mice. To visualize the diversity in these two microbial communities, principal coordinate analysis (PCoA) was done using the distance matrices and the dominant three components were plotted and ANOSIM computed[71]. Only bacterial genera represented at >/=1% relative abundance in tick samples were included in the analysis and only genera represented in >70% of the samples in either group were included. Of the 12 genera that met this criteria, only those represented in the tick samples in either Ova or PIXR group at levels 2-fold higher than in buffer sample were included (Supplemental Table 4). Additional genera observed in the buffer sample included *Neisseriacea* (8%), *Enterobacteriaceae* (21%), and *Ralstonia* (11%). Further, only those genera that also provided bona-fide amplicons in a reverse transcriptase PCR assay using fed larval RNA and genera-specific 16S rRNA primers (listed in Supplemental Table 2) were included in the final composition analysis and beta diversity computed as outlined above. The 16S rRNA amplicons obtained using genera-specific primers were sequenced to identify genera associated with larva and nymphs. Sequences obtained for each genera provided a 100% match to the following: *Staphylococcus sp.* CHNDP23 16S ribosomal RNA gene, partial sequence (accession code DQ337534.1); *Pseudomonas putida* strain D61 16S ribosomal RNA gene, partial sequence (KM488449.1); *Lysinibacillus cresolivorans* strain SC03 16S ribosomal RNA, partial sequence (NR_145635.1); *Enterococcus gallinarum* strain BHI_84-5 16S ribosomal RNA gene, partial sequence (KX674030.1); *Brevibacterium linens* strain UFC 138 16S ribosomal RNA gene, partial sequence (KU134758.1); *Acinetobacter sp.* strain S-142 16S ribosomal RNA gene, partial sequence (KX154567.1); and *Rickettsia buchneri* strain ISO7 16S ribosomal RNA, complete sequence (NR_134842.1).

**RNA probe preparation and in situ hybridization.** Forward and reverse universal primers corresponding to regions 533 and 907 nt of bacterial 16S rRNA was used to amplify bacterial 16S rRNA from fed nymphal gut and amplicons cloned into the pGEM Teasy vector (Promega) and verified by sequencing. The Plasmid was linearized to generate sense and antisense RNA by in vitro transcription using T7 or SP6 polymerase with the Hi Scribe RNA synthesis kit (New England Biolabs) and Digoxigenin-11-UTP (Roche) according to the manufacturer's instructions. Nymphal ticks fed for 48 h were dissected in MEMFA (1M MOPS, 20 mM EGTA, 10 mM MgSO$_4$, 38% Formaldehyde) fixed for one hour, dehydrated in 100% methanol and used to assess presence of bacterial RNA in guts by whole-mount in situ hybridization using Digoxigenin-labeled sense or antisense essentially as described for *Xenopus* embryos[72]. Hybridized RNA was detected with alkaline phosphatase-conjugated anti-Digoxigenin antibody (Sigma-Aldrich) and the substrate BM purple (Roche). Guts were visualized using a bright field microscope at 10 X magnification.

**Immunofluorescence microscopy.** Guts from nymphal ticks fed for ~72 h were dissected, fixed in 4% PFA for 20 min, washed in PBS /0.5% Tween20 (3 times) and blocked in PBS/0.5% Tween20, 5% Fetal Calf Serum for 1 h prior to incubation with polyclonal rabbit anti-*B. burgdorferi*-N40 (AbCam, MA) antibody and bound antibodies detected using Fluorescein isothiocyanate (FITC)-conjugated goat anti-rabbit antibody (Invitrogen) to visualize *B. burgdorferi* spirochetes. To detect PNAG the guts fixed and blocked as above were incubated with polyclonal rabbit anti-PNAG[24] (gift from Dr. Kimberley K. Jefferson) and bound antibodies detected using FITC-conjugated goat anti-rabbit antibody (Invitrogen). Nuclei were stained with 4,6-Diamidino-2-phenylindole (DAPI) in Prolong Gold Antifade mounting reagent (Invitrogen) and stained guts visualized under a Leica SP5 Fluorescence confocal microscope (Leica Microsystems) at 40X magnification.

To detect *S. aureus* PNAG in tick guts, overnight planktonically grown *S. aureus* was suspended in PBS (~2 × 10⁷ ml⁻¹) and ~5 nl injected/nymphal tick and ticks allowed to feed on PIXR or Ovalbumin-immunized mice for 48 h and processed for immunofluorescence microscopy to detect PNAG as described above.

**Assessment of the metabolome composition of the tick gut.** The pathogen-free nymphal ticks were fed on PIXR-immunized or Ovalbumin-immunized mice for

48 h to avoid excessive blood contamination that would arise in repleted ticks. The guts were dissected and pooled in groups of 10 in 50 μl of sterile PBS and stored at −80 C overnight. The next day, the tubes were spun at 1000 r.p.m. for 10 min and supernatants processed for LC-MS-based untargeted metabolomics analysis at Creative Proteomics, NY (www.creativeproteomics.com) using a UHPLC-QTOF-MS (Agilent 1290) and a UHPLC + AB QTOF 5600(ACQUITY UHPLC). 2-Chloro-L-phenylalanine (Sigma) was used as an internal standard. Metabolite extraction was done using 500 μl −20 °C pre-cold 75% methanol and 10 μl internal standard, vortexed for 1 min and centrifuge for 15 min at 12,000 r.p.m., 4 °C and the supernatant filtered using a 0.25 ml disposable syringe filter plug with 0.22 μm pore size filter, and the filtrate transferred into 2 ml vial (~0.2 ml) for subsequent analysis with ultra-performance liquid chromatography coupled with quadrupole time of flight mass spectrometry (UPLC-Q TOF). The raw file generated by MS during data acquisition were processed with XCMD for chromatographic peaks and peak identities verified using an in-house database of metabolite MS and MS/MS spectra and additionally confirmed using the public resources METLIN. The data obtained from the LC-Q TOF MS system (8043 peaks in the positive mode and 10,121 peaks in the negative mode) were then processed as described by Storey and Tibshirani[73] and filtered for denoising to remove compounds whose peak area RSD was >30%. The filtered data were then standardized by peak area normalization and processed with the SIMCA (V14.0, Umetrics AB, Umea, Sweden) for pattern recognition-multivariable analysis to obtain a PCA plot and ensure that all samples were within 95% confidence intervals. Supervised orthogonal projections to latent structures-discriminate analysis (OPLS-DA) were applied to obtain a higher level of group separation ($R^2Y$ = 0.666, 0.599 and $Q^2Y$ = 0.989, 0.999). Several sevenfold cross validation was used to estimate robustness and predictive ability of our model. The $R^2$ and $Q^2$ intercept values were 0.991 and 0.71 after 200 permutations. Differentially represented metabolites were then determined based on the value of the VIP, OPLS-DA model first principal component (threshold >1) and the $P$ value of $t$-test ($P < 0.05$) and differentially represented metabolites were analyzed by hierarchical clustering using the Euclidean distance matrix to generate the heatmaps.

**Statistical analysis and other computational tools**. The significance of the difference between the mean values of the control and experimental groups in quantitative polymerase chain assays was analyzed using the Mann–Whitney $U$-test and in all biofilm assays statistical significance was assessed using the ordinary one-way analysis of variance (ANOVA) with Dunnett's multiple comparisons test using Prism 6.0 software (GraphPad Software, USA). Protein sequence analysis and comparisons were done using proteomics tools within the ExPASy bioinformatics resource portal (Expasy.org), and SignalP 4.1 (cbs.dtu.dk). Primers for tick genes were designed using OligoPerfect™ Designer (Thermoscientific). Bacterial genera-specific primers were designed based on probeBase (probebase.csb.univie.ac.at). Nucleotide sequences matches of 16S rRNA to specific bacterial genera were confirmed by BLASTN (blast.ncbi.nlm.nih.gov). Nucleotide and protein sequences were aligned using the T-coffee multiple alignment suite (tcoffee.crg.cat or http://www.ebi.ac.uk/Tools/msa/tcoffee/).

**Data availability**. The nucleotide sequences of *pixr* (*pixr1*) and *pixr2* are deposited in the GenBank nucleotide database under accession codes KY629420 and KY865270, respectively. The 16S rDNA sequences obtained by Illumina sequencing are deposited in the GenBank Bioproject under ID PRJNA232504. The 16S rDNA analysis pertaining to this study (PIXR microbiome) can also be accessed at https://ngs.med.yale.edu/microbes/ using the guest account. The authors declare that all other relevant data supporting the findings of this study are available in this article and its Supplementary Information files, or from the corresponding authors upon request.

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

## Acknowledgements

We sincerely thank Ms Kathleen DePonte and Ms Jingyi Pan for their excellent technical assistance and Dr. Kimberly K. Jefferson of Virginia Commonwealth University, Richmond, Virginia for kindly providing the anti-PNAG antibody. We also gratefully acknowledge support from the High Performance Computing facilities and the staff of the Yale Center for Research Computing and the Yale University W.M. Keck Biotechnology Laboratory. This work was supported by NIH grants AI-49200 and AI-076705 and by a gift from the John Monsky and Jennifer Weis Monsky Lyme Disease Research Fund. The HPC computing cluster facility is funded in part by the grants RR19895 and RR029676-01. J.W.H. is funded by the Union's Seventh Framework Programme for research, technological development and demonstration under grant agreement No. 602272. E.F. is an Investigator with the Howard Hughes Medical Institute.

## Author contributions

S.N. and T.J.S. designed the experiments, performed the analysis and prepared the manuscript. N.M.A. performed the biofilm experiments and analysis. N.R. performed the bioinformatics analysis. J.C. performed tick molting experiments and analysis. M.G. and A.R. performed the electron microscopy experiments and in situ hybridization experiments respectively. M-J.W. assisted in all PCR and western blot experiments. S.D. and J.W.H. provided input during experiment design and analysis. E.F. provided resources, input towards design, analysis and prepared the manuscript. All authors contributed towards manuscript preparation.

## Additional information

**Competing interests:** The authors declare no competing financial interests.

