## [Peer Review File · Nature Communications]

Reviewers' comments:

Reviewer #1 (Remarks to the Author):

Narasimhan and colleagues present a report of a tick factor suggested to play a role in host defense against microbes. The authors use subtractive hybridization to identify a number of genes differentially expressed in *Borrelia*-infected nymphal ticks, one of which is characterized further (PIXR). The authors show PIXR expression upon feeding, and that PIXR is required for *Borrelia* colonization. Furthermore, the authors present evidence that reduction of PIXR function by immunization alters the tick metabolome.

It is important to understand mechanisms of vector-based immunity in order to combat increasingly prevalent diseases like Lyme, yet an understanding of the basic biology of vectors like ticks is lacking. Due to the inherent challenges of working with ticks, the paper therefore represents an advance by using RNAi and immunization-based approaches to inhibit PIXR function *in vivo* and assess the consequences for *Borrelia* colonization and transmission.

However, I have major concerns with the experiments and conclusions presented in Figures 4-5 of this paper.

Major concerns:

- My biggest concern with this work is the presumption that the tick midgut contains a diverse microbiome. The authors cite previously published work (Narasimhan 2013) to support the existence of a diverse midgut microbiota. I was not convinced by the data presented in Narasimhan 2013, and therefore am not convinced by the current work. The issues can be summarized as follows:

- o Larval ticks are small and the amount of bacterial-derived DNA is likely low. Low-input samples in nextgen 16S-sequencing studies are subject to being confounded by contamination, since 16S requires an amplification step. See Salter et al 2014 BMC Biology for a review of this subject, including evidence that contamination of water, DNA-isolation kits, and other reagents by low-abundance bacterial DNA can confound results. Many of the genera found in Salter et al are the same as those reported to be highly abundant in both Narasimhan 2013 and the current work (*Stenotrophomonas*, *Acinetobacter*, *Pseudomonas*, etc). Furthermore, these genera are also highly abundant and nearly ubiquitously found in the environment.

- o Several recent papers have found that endosymbionts nearly exclusively comprise the dominant bacterial constituents of the tick microbiome (See Hawlena et al, Gall et al, Rynkiewicz et al, others for work on field-collected wild ticks). However, endosymbionts like *Rickettsia* appear to be a minor fraction of the sequences found in both Narasimhan 2013 and the current study. This is perhaps most obvious in Narasimhan 2013 Fig 7B, where *Rickettsia* from field-collected ticks is dwarfed by the abundance of *Acidovorax* and others.

- o The genera reported in Narasimhan 2013 and the current study overall appear to be quite different, despite the same apparent source of ticks.

- To assuage these concerns, simple controls need to be performed. Simple negative controls for the sequencing experiments would be to sequence the water used in the study, and to wash the ticks prior to midgut dissection and sequence those samples to understand what bacteria are external. Sequences that appear in water-only controls should be excluded. However, sequences that appear in external washes may in fact also be internal. To properly establish that internal bacteria exist within the tick midgut, hybridization-based assays should be performed, such as FISH against conserved 16S rRNA sequence, using the paraffin-embedding technique used in Narasimhan 2013.

This is a crucial point. If diverse bacteria do not inhabit the tick midgut, interpretation of the experiments presented in Fig 4-5 is impossible. The results presented in Fig 1-3 and 6 stand on their own, however I do not think that Fig 1-3+6 alone warrant publication in Nature Communications.

- It is perhaps particularly telling that PIXR abrogation (and purported associated changes in the “midgut microbiota”) does not affect tick immune gene expression (Fig S4). One might expect that an expansion of environmental bacteria like *Pseudomonas* might trigger immunity (see Mulcahy et al 2011, PLoS Pathogens for an example of *Pseudomonas* impact on *Drosophila* immunity, etc).

Further comments:

- If I assume the interpretation of the authors is correct that diverse bacteria inhabit the tick, the EM evidence that bacterial biofilms exist in the tick midgut is not convincing. The authors should support their claims by specific labeling, such as wheat germ agglutinin (one example that is commonly used in the biofilm literature). Again, see Mulcahy et al which includes visualization of EPS and bacteria by IF microscopy.
- Figure 4a-ii and 4bi-ii, the panels do not look qualitatively different. Recommend quantification.
- The authors should include “dysbiosed” or antibiotic-treated ticks as negative controls demonstrating what the absence of “biofilms” might look like in comparison.
- I am not aware of differences between gram-positive and gram-negative biofilms that are conserved broadly across genera or species. How do the authors propose PIXR exhibits specificity for gram-positive biofilms?

Reviewer #2 (Remarks to the Author):

The manuscript by Narasimhan et al, entitled “PIXR, a secreted *Ixodes scapularis* protein, modulates the gut milieu in favor of *Borrelia burgdorferi* colonization of the tick” follows up a previous report of the group (Narasimhan et al., *Cell Host & Microbe* 2014) in which the authors demonstrated that colonization of spirochetes in the tick gut is not merely affected by tick-pathogen interactions but involves also a complex interactome with gut dwelling microbiota. In the current work, the authors propose the role of one reeler-related protein termed PIXR in the spirochete colonization of *I. scapularis* midgut. The gene encoding PIXR was found among others to be up-regulated by *Borrelia burgdorferi* (Bb) presence in the nymphal midgut by the method of subtractive hybridization using *Borrelia*-infected and clean nymphal guts as the Tester and Driver probes, respectively. In further the authors demonstrated by a series of more-than-less indirect evidences that PIXR modulate spirochete colonization of the nymphal midgut by manipulating the present microbiome and resulting composition of metabolites via affecting the biofilm formation of Gram-positive bacteria. In contrast to the phylogenetically distantly-related reeler proteins of other arthropods, PIXR did not exert any antimicrobial activity. The study overall is interesting as it provides new ideas how tick microbiome may affect the spirochete survival in the tick vector. On the other hand, I find the submitted evidences rather unconvincing and indirect to convict PIXR to be the molecule responsible for the observed effect.

The main achieved results could be briefly enumerated as follows:

- In accord with the subtractive hybridization result, qRT-PCR analysis confirmed that PIXR

expression is up-regulated in the midguts of Bb-infected nymphs

- PIXR is mainly expressed in midguts and mRNA levels increase during feeding
- RNAi silencing of PIXR resulted in about 50% reduction of PIXR mRNA level in nymphal midgut, had no effect on nymphal engorgement weight but led to significant reduction of Bb present in the nymphal midgut
- Vaccination of mice with recombinant PIXR reduced Bb burden in larvae and impaired larval molting but did not seem to affect Borrelia transmission from infected nymph to the mice
- Recombinant PIXR did not seem to exert antimicrobial activity as it did not affect in-vitro growth of Gram-positive, Gram-negative bacteria and Borrelia
- Electron microscopy of gut sections from nymphs fed on PIXR-immunized mice suggested that blocking of PIXR affects formation of bacterial biofilms. The role of PIXR in inhibition of biofilm formation was demonstrated in vitro for Gram-positive bacteria *Staphylococcus aureus* but not for Gram-negative bacteria *Pseudomonas aeruginosa*
- Feeding larvae on mice immunized with rPIXR had no effect on expression of selected tick immune genes
 - *I. scapularis* larvae fed on rPIXR-immunized mice had different microbiome (based on 16S analysis compared to the control immunized with ovalbumin.
 - The increase in the representation of the genera *Brevibacterium* in larvae fed on rPIXR immunized mice was corroborated by in vitro test showing inhibition of biofilm formation for *Brevibacterium casei*.
 - The HPLC/MS metabolome analysis of gut homogenates from partially fed nymphs revealed significant differences in metabolite composition between ticks fed on rPIXR and ovalbumin immunized mice
 - Expression of some *Borrelia*-specific genes were different in larvae fed on ovalbumin and rPIXR immunized mice

Major points:

1. One of the main concern I have about this work is that the proposed effect of PIXR was demonstrated either upon RNAi silencing of PIXR gene in *I. scapularis* nymphs or upon larval or nymphal feeding on rPIXR-immunized mice. In no case a similar phenotype was by obtained by both independent approaches used to abrogate the PIXR function which would corroborate the obtained results. The mixed, mutually unrelated results reported either for larval or nymphal stages make the story rather chaotic and hardly comprehensible especially for someone out of the field.

There is much uncertainty about RNAi in nymphs. The authors show that RNAi in nymphs was successful in non-infected nymphs (Figure 3B) but it did not work in Bb-infected nymphs (data not shown, lines 152, 153). This is quite strange and probably the RNAi experiments in nymphs needs to be optimized and repeated

Having that, an additional RNAi experiment should be performed to see the nymphal gut ultrastructure upon PIXR-RNAi.

Even though quite demanding, it would be also beneficial for the impact of the work to see the microbiome/metabolome differences also in the midguts from nymphs upon PIXR RNAi-silencing

2. Are the author sure that only the described PIXR is responsible for the described effects?

BLAST search using the ISCW005667 gene sequence across the *I. scapularis* ESTs or whole genome shotgun sequences available at NCBI GenBank, reveal existence of a whole array of highly similar reeler-related genes. The same result was obtained by the BLAST search in the available transcriptomes from the closely related tick species *Ixodes ricinus*. This fact should be confessed and demonstrated by presenting more tick reeler-related genes in the multiple sequence alignment in the Figure 1.

In light of this, it is not clear at all that targeting the PIXR in larvae by specific antibodies and in the nymphs by RNAi really hit the same gene/protein.

The Western Blot results shown in the Figure 2D and 2E detect the PIXR in nymphal midgut homogenate as a clear double band. To my opinion, it may reflect the presence of at least two closely related PIXR isoforms rather than the suggested post-translational modification of one protein. In order to unambiguously answer this question, another PIXR-RNAi silencing experiment in nymphs should be performed with following Western blot analysis of nymphal midgut homogenates RNAi. In case this experiment demonstrate two different PIXRs in nymphal midgut, than the redundancy may explain the lack of expected phenotypes.

3. In testing the potential antimicrobial activity of rPIXR against various bacteria (Supplemental Fig. 2), no positive control with any specific antibacterial peptide (defensin, cecropin or similar) or at least antibiotics was used. It is therefore impossible to judge whether the grow assays worked as performed.

4. The results obtained by electron microscopy shown in Figure 4 A, B are at first look not much convincing. I realize that it is impossible to make a judgment without examining and comparing dozens of images of midgut sections from the control and experimental group. Therefore accepting this evidence and interpretation of the obtained results can be based only on the high credibility of the authors' team.

A timepoint of 48h for EM and metabolome examination of nymphal midguts selected by the authors was reasoned to avoid the excessive contamination with the blood meal. On the other hand, it should be taken into consideration that at this time point, the experimental group may comprise a mixture of nymphs at different feeding progress with no chance to distinguish between male and female nymphs.

How does the biofilm look in unfed ticks, is it formed at all?

5. A scapularisin type of defensin has been identified by subtractive hybridization to be upregulated in Bb-infected ticks. This is an interesting finding worth of further investigation. However, the authors get over this with a statement (lines 108-110) that the "scapularisin peptide has not been detected" with a reference to the review article by Sonenshine and Hynes, 2008. To my knowledge, there is no information about this particular scapularisin in this review. Instead the authors may better assign the identified scapularisin to one of 25 scapularisins described in I. scapularis (Wang & Zhu, 2011, Devel. Comp. Immunol, 351128-34).

Minor points (typos):

line 113 "Pediculis" change to "Pediculus"

line 124 "IXodes" change to "Ixodes"

line 130-131 "larvae that fed on clean or B. burgdorferi -infected larvae" change to "larvae that fed on clean or B. burgdorferi-infected mice"

line 473 "Syber Green" change to "Sybr Green"

line 576 "S. aureus" change to "S. aureus"

line 577 "P. aeruginosa" change to "P. aeruginosa", "B. casei" change to "B. casei"

Reviewer #3 (Remarks to the Author):

What are the major claims of the paper?

Following on previous work by these authors that demonstrated that tick gut microbial composition influenced *Borrelia burgdorferi* (the causative agent of Lyme disease) colonization, this study provides compelling evidence for the role of a protein with a Reeler domain (PIXR) in the facilitation of colonization of ticks by *B. burgdorferi*. They provide evidence to show that PIXR, is a secreted gut protein and is upregulated in the presence of *B. burgdorferi* and that when PIXR is inactivated, using both RNAi and by exposing to anti-PIXR antibodies, that *B. burgdorferi* is less effective at colonizing ticks. The mechanisms by which PIXR impacts *B. burgdorferi* colonization was investigated and a number of potential direct and indirect effects were observed and/or proposed, including impacts on Peritrophic Membrane structure, bacterial biofilm formation (possibly in a Gram-status specific manner), related impacts on gut microbiome composition and metabolites.

Are they novel and will they be of interest to others in the community and the wider field?

This study also expands on the growing body of literature related to Reeler domain proteins (such as Noduler) that play a key role in invertebrate response to microbial infection/colonization. In this case however the protein does not seem to have a bactericidal function, rather it appears to exert selection on the microbiome, reducing biofilm formation/expansion/ of Gram positive bacteria in a gut microbiome that is predominated by Gram negative bacteria. The selective nature of this interaction is fascinating and no doubt underlain by complex molecular recognition processes that are not discussed (nor within the scope of this study). Given the homology of PIXR within many other invertebrates, this study should be of broad appeal to many working in vector borne diseases, invertebrate ecology, biocontrol and host-microbe interactions in general.

If the conclusions are not original, it would be helpful if you could provide relevant references.

I believe the conclusions are original and in most cases supported by the data presented.

Is the work convincing, and if not, what further evidence would be required to strengthen the conclusions?

I had some difficulty in navigating the manuscript, although it is generally well written the results and discussion intertwine both the direct and indirect mechanisms by which PIXR could exert the observed phenotype. The manuscript also investigates PIXR impacts at both larval and nymph stage and in several sections it is not clear which growth stage is referred to – this is especially important given the timing of the development *Borrelia*-tick associations in nature (i.e. occurring at larval and not nymph stage). For this reason I think the manuscript would benefit from a summary figure of a conceptual model that highlights the potential mechanisms and evidence from both stages. For example the authors describe alterations to the Peritrophic Membrane as observed using electron microscopy and Schiff staining, as a Reeler domain protein it is very possible that PIXR is involved in regulating the composition of the PM (in addition to bacterial EPS) and that this impacts colonization by *B. burgdorferi*. And while I tend to agree with the line of reasoning that the impact of PIXR is through selective suppression of Gram positive growth in biofilms, the authors previous work demonstrated that dysbiotic gut microbiota resulted in significant changes in expression of the STAT transcription factor – in this case STAT expression was not altered when PIXR was presumably abrogated through feeding on fed on PIXR-immunized mice. Other responses to dysbiotic gut microbiota observed in the previous study but not observed here include the decrease in peritrophin (component of the PM). This suggests that two or more separate mechanisms are at play. So while the combination of multiple lines of evidence makes this a very strong manuscript, I still think it would have benefited from an experiment to de-couple the direct and indirect effects of PIXR. For example, like in the authors previous manuscript, a suggestion would be to use an antibiotic treatment. In this case to suppress Gram-positive bacteria in the ticks fed on PIXR-immunized mice that showed increased *Brevibacter* – if the

mechanism promoting *B. burgdorferi* colonization is related to repressing Gram-positive biofilm growth, then using Gram-positive specific antibiotics in this case would confirm this by reducing *Brevibacter* and the other Gram-positives and increasing *B. burgdorferi* in the ticks fed on PIXR-immunized mice.

While the majority of the manuscript is string and evidence based, the connection to the metabolite composition and its role in serving as a cue for *B. burgdorferi* colonization is the weakest link. I admire the lengths the authors went to, to address this important question, but it was not nailed down in anyway and a large part of the discussion and the overall conclusions are based on what remains speculation. I am sure they are following this up in current work but those claims need to be tempered until the conclusive data are acquired.

As a curiosity, how does the gut microbiome of laboratory raised ticks compare to those in nature – obviously this would be important given the hypothesis here that the molecular cues for *B. burgdorferi* colonization are derived from the gut microbial metabolites. I know this has been addressed in previous manuscripts including by these authors but a discussion of this would be valuable.

Specific comments:

L330: What does gut genes refer to? Host genes or host and microbiome? Suggest rewording.

L331: I would describe the microbiome as more than a “new correlate”.

L369-373: There is so much other worthy detail that this connection to copper being toxic and *Cupriavidus* purifying the environment is just speculation and not worth including in my opinion without data. Also *Cupriavidus* is not ‘unique’ in that aspect.

L389: C-di-GMP upregulation is interesting for many reason, including its connection to EPS production in response to nitric oxide – that could be an important mechanism in *B. burgdorferi* establishment – maybe NO is necessary?

L393-398: The metabolite data is still preliminary without identification – all this says is that metabolites are different, and there are some upregulated *Borrelia* genes that have been related to tick colonization (but that have not been shown to be essential for that process yet) but there is no way to assign any sort of causality at this point. Again I’m sure the authors are going there but the evidence in this paper does not support this so I suggest they ease off on the length of discussion around this topic and the emphasis in the abstract/discussion.

Response to Reviewers comments

We thank the reviewers for the careful review. We have heeded their suggestions and have performed additional experiments to further clarify and strengthen the findings. Our point-by-point response (in italics) to their specific comments (underlined) are provided below.

Reviewer #1 (Remarks to the Author):

Narasimhan and colleagues present a report of a tick factor suggested to play a role in host defense against microbes. The authors use subtractive hybridization to identify a number of genes differentially expressed in *Borrelia*-infected nymphal ticks, one of which is characterized further (PIXR). The authors show PIXR expression upon feeding, and that PIXR is required for *Borrelia* colonization. Furthermore, the authors present evidence that reduction of PIXR function by immunization alters the tick metabolome. It is important to understand mechanisms of vector-based immunity in order to combat increasingly prevalent diseases like Lyme, yet an understanding of the basic biology of vectors like ticks is lacking. Due to the inherent challenges of working with ticks, the paper therefore represents an advance by using RNAi and immunization-based approaches to inhibit PIXR function in vivo and assess the consequences for *Borrelia* colonization and transmission. However, I have major concerns with the experiments and conclusions presented in Figures 4-5 of this paper.

Major concerns:

My biggest concern with this work is the presumption that the tick midgut contains a diverse microbiome. The authors cite previously published work (Narasimhan 2013) to support the existence of a diverse midgut microbiota. I was not convinced by the data presented in Narasimhan 2013, and therefore am not convinced by the current work. The issues can be summarized as follows:

Larval ticks are small and the amount of bacterial-derived DNA is likely low. Low-input samples in nextgen 16S-sequencing studies are subject to being confounded by contamination, since 16S requires an amplification step. See Salter et al 2014 BMC Biology for a review of this subject, including evidence that contamination of water, DNA-isolation kits, and other reagents by low-abundance bacterial DNA can confound results. Many of the genera found in Salter et al are the same as those reported to be highly abundant in both Narasimhan 2013 and the current work (Stenotrophomonas, Acinetobacter, Pseudomonas, etc). Furthermore, these genera are also highly abundant and nearly ubiquitously found in the environment.

We agree with the reviewer that environmental contamination might inflate the complexity of the tick microbiome. We have given this point considerable and careful thought. We have now conducted the buffer controls suggested by the reviewer and conducted a mock extraction of DNA under identical conditions using water, buffers and kits utilized for the experimental samples followed by Illumina MiSeq analysis of

16S rRNA reads to determine bacterial genera that are likely present in the reagents and kits utilized for DNA extraction, amplification and purification steps. We obtained about 8000 reads from this control (compared to an average 35,000 reads in each of the larval samples) sample. Predominant bacteria in the mock sample were Enterobacteriaceae, Ralstonia, Neisseriaceae, and Proteus. Some genera that were observed in larval samples were also observed in the mock sample (Supplemental Table 4). We have now revised our analysis to only include bacterial genera that were: **A.** represented in greater than 70 % of the tick samples; **B.** represented at greater than 1 % in abundance in the PIXR or Ovalbumin group; **C.** that were at least 2-fold greater in abundance in larval samples in the PIXR or Ovalbumin group compared to that in buffer control and **D.** that could also be detected by qRT-PCR using genera-specific 16S rRNA primers. Using these additional criteria to define tick microbiota we observe that *I. scapularis* ticks are associated with seven bacterial genera (Figure 7). Genera such as *Brevibacteria*, *Acinetobacter* and *Pseudomonas* that the reviewer has suggested could be environment contaminants continue to emerge as tick-associated bacteria. Bacteria represented commonly in the environment and even perhaps on murine skin such as *Staphylococcus* might become associated with the tick. Understanding the functional role of each of these tick-associated bacteria in tick physiology will be critical to our evolving understanding of tick microbiota. We acknowledge this in the revised manuscript but are also very confident about the extensively detailed analysis that we have performed.

Several recent papers have found that endosymbionts nearly exclusively comprise the dominant bacterial constituents of the tick microbiome (See Hawlena et al, Gall et al, Rynkiewicz et al, others for work on field-collected wild ticks). However, endosymbionts like Rickettsia appear to be a minor fraction of the sequences found in both Narasimhan 2013 and the current study. This is perhaps most obvious in Narasimhan 2013 Fig 7B, where Rickettsia from field-collected ticks is dwarfed by the abundance of Acidovorax and others.

We agree that there are several publications (Clayton et al., 2015; Gall et al., 2016) including the ones that the reviewer has brought to our attention, that demonstrate significantly less diversity in *Ixodes* ticks, but several published reports on the microbiome of *I. scapularis* and other tick species also observe increased diversity (Andreotti et al., 2011; Budachetri et al., 2014; Budachetri et al., 2016; Carpi et al., 2011; Kurilshikov et al., 2015; Nakao et al., 2013; Trout Fryxell and DeBruyn, 2016). We agree with the reviewer that controls such as the buffer and additional controls proposed by Salter et al (Salter et al., 2014) would be critical to discern specific tick-associated bacteria. Several factors including tick species, gender, geographic location (Van Treuren et al., 2015), developmental status, time since molting and feeding (Menchaca et al., 2013) and infection status (Abraham et al., 2017) are also expected to have an impact on the composition of the microbiome. Further, differences in the sequencing platforms, and differences in the 16S variable regions targeted for amplification may also contribute to the contrasting observations on the microbial diversity. Visualization of specific bacteria in the tick in conjunction with a functional role for specific bacterial associations might be essential to determine bona-fide members of the tick microbiome. At this juncture, we infer that the bacterial genera reported in this manuscript are frequently associated with lab-reared and murine host-fed *I. scapularis* larval ticks. We have indicated this in the revised text and rephrased to note that these bacterial genera are tick-associated microbiota.

Techniques to address vertebrate and invertebrate microbiomes continue to evolve and shape our understanding of various microbiomes including that of ticks. This field is rapidly evolving and data using both field-collected and laboratory ticks will contribute to a better understanding of these concerns.

The genera reported in Narasimhan 2013 and the current study overall appear to be quite different, despite the same apparent source of ticks.

This is indeed an important point and could have been highlighted in the earlier Discussion. Importantly, the previous study utilized the 454-sequencing platform and the current study has utilized the Illumina MiSeq platform for the 16S rDNA sequencing. Comparing the composition of control fed larval ticks from the previous study (Narasimhan et al., 2014) with the control fed larval ticks from this study we observe the following common genera (Stenotrophomonas, Acinetobacter, Enterococcus, Lysinibacillus, Rickettsia, Chryseobacterium, Corynebacterium, Acidovorax, Brevibacterium, Staphylococcus). The revised analysis has included additional inclusion and exclusion criteria as outlined above, that was not utilized in the earlier study, and therefore presents differences in the final assessment of the microbiome composition and we have discussed this in the revised manuscript. Additionally, tick guts potentially harbor bacteria that are maternally inherited transovarially, transovum as well as those that the tick is exposed to in the environment and the interplay and competition between these bacteria might determine the tick-associated microbiome composition. While the larval and nymphal ticks are primarily reared in our laboratory-they are an outbred population raised from adult ticks either obtained from the Connecticut Agricultural Experiment Station, or from field-caught adults from endemic areas in Connecticut, or from the Oklahoma State University, Noble Research Centre. We therefore speculate that batch variations between the ticks used in this study and in the previous study might also account for the differences in the microbial compositions. Within each study the same batch of ticks were used and differences observed between the control and experimental groups of larvae or nymphs within each study is valid and conforms to experimental rigour. In both studies, the ticks were surface sterilized in 70 % ethanol 10 minutes, followed by two sterile water washes. The microscope stage for dissection and slides used for dissection were sterilized and soaked in 70 % ethanol prior to use. We have now included this in the Discussion and Materials and Methods section to clarify the variations in the compositions of larval ticks in this and the previous study.

To assuage these concerns, simple controls need to be performed. Simple negative controls for the sequencing experiments would be to sequence the water used in the study, and to wash the ticks prior to midgut dissection and sequence those samples to understand what bacteria are external. Sequences that appear in water-only controls should be excluded. However, sequences that appear in external washes may in fact also be internal. To properly establish that internal bacteria exist within the tick midgut, hybridization-based assays should be performed, such as FISH against conserved 16S rRNA sequence, using the paraffin-embedding technique used in Narasimhan 2013. This is a crucial point. If diverse bacteria do not inhabit the tick midgut, interpretation of the experiments presented in Fig 4-5 is impossible. The results presented in Fig 1-3 and 6 stand on their own, however I do not think that Fig 1-3+6 alone warrant publication in Nature Communications.

We agree with these thoughtful suggestions to strengthen the findings. As noted above, we have performed the water/buffer control suggested by the reviewer and present a revised analysis of tick-associated microbiota (Figure 7). We agree with the reviewer that visualization of bacteria within the gut would help further validate the findings. Paraffin embedding followed by FISH to visualize bacteria has posed technical hurdles. There is one report of FISH to visualize *B. burgdorferi* a bacterium known to colonize the tick gut (Hammer et al., 2001) and of tick ovaries for specific endosymbionts (Klyachko et al., 2007). Attempts to utilize FISH to visualize gut microbiota provided sub-optimal results. Inherent auto-fluorescence of tick guts made it difficult to unambiguously determine bacterial presence. We took an *in situ* hybridization approach using universal bacterial 16S rRNA probes and visualization by bright field microscopy (Khokha et al., 2002). While we could not perform this with larval ticks due to the small size, we were successful in examining nymphal ticks. This approach circumvented auto-fluorescence issues and also allowed enhanced visualization of hybridizing probes. This showed that bacteria were indeed present in the gut diverticula (Figure 7). Genera-specific probes for *Acinetobacter*, *Brevibacterium*, and *Pseudomonas* to visualize specific genera within the guts, also provided signal and is included (Supplemental Figure 6). Technical advancements will likely improve our ability to better visualize tick gut bacteria.

Further, we surface sterilized several batches of larval and nymphal ticks and purified RNA from whole larvae and from dissected guts of nymphs. We then performed quantitative RT-PCR assays using genera-specific primers to determine if RNA representing these bacterial genera were detectable in nymphal guts and larvae indicative of viable bacteria within the ticks (Supplemental Figure 6). Our results suggest that these bacterial genera were indeed represented in the ticks. The genera-specific amplicons were sequenced to confirm that they indeed amplified specific genera and we are able to submit the sequences to Genbank if warranted.

It is perhaps particularly telling that PIXR abrogation (and purported associated changes in the “midgut microbiota”) does not affect tick immune gene expression (Fig S4). One might expect that an expansion of environmental bacteria like *Pseudomonas* might trigger immunity (see Mulcahy et al 2011, PLoS Pathogens for an example of *Pseudomonas* impact on *Drosophila* immunity, etc).

While we did not observe significant changes at the transcript levels for *stat*, *duox*, and *nos*, we recognize that changes could be happening at the protein level and until we have antibodies for all these tick immune-response proteins, we cannot conclude that immune responses were not altered. Nevertheless, this is indeed a valid point and in the revised manuscript we have addressed the expression profiles of several additional immune response genes. Not all the immune response pathways and their effectors are fully annotated in the *I. scapularis* genome. Based on identified arthropod homologs (Palmer and Jiggins, 2015; Smith and Pal, 2014) we examined key components of the IMD, JNK and Toll pathways. We observed that transcript levels of *relish*, a component of the IMD (Immunodeficiency) pathway that responds to gram-negative bacteria, were not altered while transcript levels of *bsk*, a component of the JNK (Jun amino-terminal kinase) signaling pathway involved in development and response to microbial challenge, stress, and epithelial injury (Delaney et al., 2006), were decreased upon PIXR abrogation (Figure 6). We note that the tick gut is predominantly associated with gram-negative bacteria and ticks have evolutionarily lost key components of the IMD pathway (Palmer and Jiggins, 2015) that in other

arthropods represent a critical sensor of gram-negative bacteria (Lemaitre, 2004).

Ticks do have the major elements of a Toll-like pathway to sense gram-positive bacteria such as *Brevibacterium* observed to increase upon PIXR abrogation. Dorsal, the key transcription factor in the Toll pathway, was increased upon PIXR abrogation (Figure 6). Not much is known of tick effector molecules of immune response pathways. At the transcript level, we examined the annotated antibacterial peptide molecules, Scapularisin1 and 5 (Palmer and Jiggins, 2015; Wang and Zhu, 2011) potentially representing effector molecules of the Toll-like pathway and found that scapularisin5 was significantly decreased (Supplemental Figure 4) - suggesting an immune suppressive effect on the Toll-like pathway upon PIXR abrogation. A putative secreted peptidoglycan recognition protein (Palmer and Jiggins, 2015) and DAE (Domesticated amidase effector) an amidase invoked in controlling *B. burgdorferi* (Chou et al., 2014; Smith et al., 2016) was also significantly decreased upon PIXR abrogation (Supplemental Figure 4). Biofilm components have a complex interaction with host immune responses depending on the type of bacteria and the tissue where the biofilms are localized (Watters et al., 2016). Several reports have shown that the excreted polysaccharides of biofilm have immune suppressive effects in the mammalian host (Bylund et al., 2006; Jensen et al., 1993; Murofushi et al., 2015). Biofilm increase upon PIXR abrogation might potentially suppress the immune responses of the tick. However, until the *I. scapularis* immunome is comprehensively characterized it would be premature to conclude that PIXR abrogation has a global immunosuppressive effect. We have now included the new data and the corresponding Discussion in the revised manuscript.

Further comments:

• If I assume the interpretation of the authors is correct that diverse bacteria inhabit the tick, the EM evidence that bacterial biofilms exist in the tick midgut is not convincing. The authors should support their claims by specific labeling, such as wheat germ agglutinin (one example that is commonly used in the biofilm literature). Again, see Mulcahy et al which includes visualization of EPS and bacteria by IF microscopy.

*Wheat germ agglutinin unfortunately provides a very high background potentially due to its binding to sialylated glycoproteins in the tick. We therefore used rabbit polyclonal antibody against Poly N-acetyl glucosamine (PNAG), a component of many gram-positive biofilms, to visualize biofilms in tick guts (Cywes-Bentley et al., 2013). Control guts showed less PNAG signal compared to PIXR-knockdown guts (Figure 5). We also injected *S. aureus* that readily forms biofilms into tick guts and assessed biofilm formation in ticks that had fed on Ova or PIXR-immunized mice to additionally demonstrate that in the absence of PIXR biofilm formation is increased (Figure 5).*

Figure 4a-ii and 4bi-ii, the panels do not look qualitatively different. Recommend quantification. The authors should include “dysbiosed” or antibiotic-treated ticks as negative controls demonstrating what the absence of “biofilms” might look like in comparison.

We have now quantified the spirochetes in the TEM panels and is now included (Figure 4). We have now included nymphal ticks that were allowed to feed on gentamicin-injected and PIXR-immunized mice. The biofilm-like structures were

considerably diminished suggesting that these are likely bacterial biofilms (Figure 4).

I am not aware of differences between gram-positive and gram-negative biofilms that are conserved broadly across genera or species. How do the authors propose PIXR exhibits specificity for gram-positive biofilms?

We do not yet know the mechanism by which PIXR inhibits gram positive bacterial biofilms. It is likely that PIXR binds to components of gram positive peptidoglycan that are not readily accessible in gram negative bacteria and require further investigation. This is now noted in the revised Discussion section and will be addressed in future efforts.

Reviewer #2 (Remarks to the Author):

The manuscript by Narasimhan et al, entitled “PIXR, a secreted Ixodes scapularis protein, modulates the gut milieu in favor of Borrelia burgdorferi colonization of the tick” follows up a previous report of the group (Narasimhan et al., Cell Host & Microbe 2014) in which the authors demonstrated that colonization of spirochetes in the tick gut is not merely affected by tick-pathogen interactions but involves also a complex interactome with gut dwelling microbiota. In the current work, the authors propose the role of one reeler-related protein termed PIXR in the spirochete colonization of I. scapularis midgut. The gene encoding PIXR was found among others to be up-regulated by Borrelia burgdorferi (Bb) presence in the nymphal midgut by the method of subtractive hybridization using Borrelia-infected and clean nymphal guts as the Tester and Driver probes, respectively. In further the authors demonstrated by a series of more-than-less indirect evidences that PIXR modulate spirochete colonization of the nymphal midgut by manipulating the present microbiome and resulting composition of metabolites via affecting the biofilm formation of Gram-positive bacteria. In contrast to the phylogenetically distantly-related reeler proteins of other arthropods, PIXR did not exert any antimicrobial activity. The study overall is interesting as it provides new ideas how tick microbiome may affect the spirochete survival in the tick vector. On the other hand, I find the submitted evidences rather unconvincing and indirect to convict PIXR to be the molecule responsible for the observed effect.

The main achieved results could be briefly enumerated as follows:

- In accord with the subtractive hybridization result, qRT-PCR analysis confirmed that PIXR expression is up-regulated in the midguts of Bb-infected nymphs
- PIXR is mainly expressed in midguts and mRNA levels increase during feeding
- RNAi silencing of PIXR resulted in about 50% reduction of PIXR mRNA level in nymphal midgut, had no effect on nymphal engorgement weight but led to significant reduction of Bb present in the nymphal midgut
- Vaccination of mice with recombinant PIXR reduced Bb burden in larvae and impaired larval molting but did not seem to affect Borrelia transmission from infected nymph to the mice
- Recombinant PIXR did not seem to exert antimicrobial activity as it did not affect in-vitro growth of Gram-positive, Gram-negative bacteria and Borrelia
- Electron microscopy of gut sections from nymphs fed on PIXR-immunized mice suggested that blocking of PIXR affects formation of bacterial biofilms. The role of PIXR in inhibition of biofilm formation was demonstrated in vitro for Gram-positive bacteria Staphylococcus aureus but not for Gram-negative bacteria Pseudomonas aeruginosa

- Feeding larvae on mice immunized with rPIXR had no effect on expression of selected tick immune genes
- *I. scapularis* larvae fed on rPIXR-immunized mice had different microbiome (based on 16S analysis compared to the control immunized with ovalbumin).
- The increase in the representation of the genera *Brevibacterium* in larvae fed on rPIXR immunized mice was corroborated by in vitro test showing inhibition of biofilm formation for *Brevibacterium casei*.
- The HPLC/MS metabolome analysis of gut homogenates from partially fed nymphs revealed significant differences in metabolite composition between ticks fed on rPIXR and ovalbumin immunized mice
- Expression of some *Borrelia*-specific genes were different in larvae fed on ovalbumin and rPIXR immunized mice

Major points:

1. One of the main concern I have about this work is that the proposed effect of PIXR was demonstrated either upon RNAi silencing of PIXR gene in *I. scapularis* nymphs or upon larval or nymphal feeding on rPIXR-immunized mice. In no case a similar phenotype was by obtained by both independent approaches used to abrogate the PIXR function which would corroborate the obtained results. The mixed, mutually unrelated results reported either for larval or nymphal stages make the story rather chaotic and hardly comprehensible especially for someone out of the field.

*We regret this confusion. We wish to clarify that both nymphal ticks and larval ticks provided the same phenotype –i.e., decreased colonization of ticks by *B. burgdorferi* upon abrogation of PIXR by RNAi in nymphs (Figure 3D-E) and anti-PIXR antibodies in nymphal ticks (Figure 3F). RNA interference is not feasible in larval ticks due to their small size and increased mortality upon injection trauma. Hence RNAi was performed only in nymphal ticks. Thus, PIXR abrogation by anti PIXR antibodies in larval ticks phenocopied (Fig 3H) the observation in nymphal ticks upon RNAi-mediated knockdown of PIXR and in nymphal ticks upon abrogation of PIXR by anti-PIXR antibodies. Transmission of *B. burgdorferi* from nymphal ticks to the murine host was not affected by RNA interference or by anti-PIXR antibodies (Supplemental Figure 2)-suggesting a redundant function for PIXR during migration of spirochetes from the tick to the murine host. Larval ticks are not capable of transmitting *B. burgdorferi*. We have now clarified this in the revised manuscript. SEM, TEM, confocal microscopy and in situ hybridizations were performed using the nymphal stage since the larval ticks are not amenable to these procedures due to the small size. Further, in the interest of clarity, and to provide an overview of the highlights of our findings using the two stages, we have included a graphical summary of the work (Figure 9).*

There is much uncertainty about RNAi in nymphs. The authors show that RNAi in nymphs was successful in non-infected nymphs (Figure 3B) but it did not work in Bb-infected nymphs (data not shown, lines 152, 153). This is quite strange and probably the RNAi experiments in nymphs needs to be optimized and repeated

*RNAi-mediated abrogation of PIXR is efficient in uninfected nymphal ticks suggesting that the dsRNA constructs and administration route were robust. Thus we are able to address by RNAi the role of PIXR in facilitating *B. burgdorferi* colonization*

of the tick gut as the spirochetes migrate from the murine host to the tick. We have now clarified that RNAi worked sub-optimally only in B. burgdorferi-infected nymphs. Transcript levels of PIXR in B. burgdorferi-infected nymphs are increased and therefore RNAi-mediated decrease is less efficient in reducing PIXR transcripts in these ticks. Therefore, RNAi approach was not utilized to address the role of PIXR in B. burgdorferi transmission from the tick to the host-instead an immunization approach was utilized.

Having that, an additional RNAi experiment should be performed to see the nymphal gut ultrastructure upon PIXR-RNAi.

We agree and have now performed an RNAi-mediated knockdown of PIXR in uninfected nymphal ticks, placed them on B. burgdorferi-infected mice and included the transmission electron microscopy of tick guts (Figure 4).

Even though quite demanding, it would be also beneficial for the impact of the work to see the microbiome/metabolome differences also in the midguts from nymphs upon PIXR RNAi-silencing

We have focused on the larval tick microbiome as it is the stage most relevant to B. burgdorferi acquisition and colonization. While the microbiome analysis was not done in the nymphal ticks, we have assessed specific genera (shown to be modulated in larval ticks) by QRT-PCR of nymphal tick guts (Figure 7 and Supplemental Figure 6) and show that these bacteria were also represented in nymphal guts and altered upon PIXR abrogation.

2. Are the author sure that only the described PIXR is responsible for the described effects?

BLAST search using the ISCW005667 gene sequence across the I. scapularis ESTs or whole genome shotgun sequences available at NCBI GenBank, reveal existence of a whole array of highly similar reeler-related genes. The same result was obtained by the BLAST search in the available transcriptomes from the closely related tick species Ixodes ricinus. This fact should be confessed and demonstrated by presenting more tick reeler-related genes in the multiple sequence alignment in the Figure 1.

We agree with the reviewer and have now performed a careful analysis to correct this oversight. Indeed, there are at least seven paralogs of PIXR in the tick genome and a multiple alignment of the paralogs shows the extent of similarity between the paralogs (Figure 1). PIXR1 (PIXR), identified in this report, is not identical to the annotated PIXR paralogs including ISCW005667 as revealed by the analysis of the updated annotations of the I. scapularis genome. PIXR sequence is now deposited in the Genbank (KY629420). We have now revised the figure to highlight homologs of PIXR in I. ricinus, a tick closely related to I. scapularis and in other arthropods (Figure 1) as suggested by the reviewer.

In light of this, it is not clear at all that targeting the PIXR in larvae by specific antibodies and in the nymphs by RNAi really hit the same gene/protein.

We agree that we must clarify which of these PIXRs might contribute to the

observed phenotype. We have now performed a comprehensive analysis of the seven paralogs). We performed an expression analysis by quantitative RT-PCR and results of the expression analysis suggest that only PIXR 1, 2, 3, 4 and 5 are expressed in larval and nymphal ticks (Supplementary Figure 1). PIXR 6 and 7 could not be detected in either stage. PIXR 1 (PIXR) and PIXR 2 and 3 are the only paralogs with a predicted signal sequence suggesting that the effect on the microbiome composition is likely limited to PIXR 1, 2 and/or 3. PIXR 1, 2 and 3 encode for predicted proteins of ~ 15, 11 and 16 kDa. The anti PIXR antibody recognizes a doublet at ~ 15 kDa in both larval and nymphal ticks. To further confirm this, we excised the doublet band recognized by the anti-PIXR antibody from larval extracts and nymphal midguts and sequenced it by LC-MS/MS mass spectroscopy. We could only detect peptides that corresponded to PIXR 1 and PIXR 2. While PIXR is predicted to have 2 O-glycosylation sites, PIXR 2 has 6 O-glycosylation sites. It is likely that these post-translational modifications might provide a higher molecular mass for PIXR2. Recombinant PIXR generated in the *Drosophila* expression system was not glycosylated as seen by performing a Periodic acid-Schiff's glycoprotein staining. Assessing the glycosylation status of PIXR1/2 by deglycosidation of gut protein extracts was not optimal. The two paralogs are ~82% identical at the nucleotide level and dsRNA that could selectively target one of these paralogs was not feasible. The dsRNA utilized to target PIXR decreased the expression of PIXR 1 and 2 as seen by western blot (Figure 3) and also decreased *pixr2* transcripts (Supplemental Figure 1). We therefore speculate that PIXR1 and 2 of the PIXR family are likely both involved in maintaining microbial homeostasis. PIXR3 protein is likely expressed at very low levels. PIXR paralogs (PIXR 4 and 5) that are likely cytosolic proteins might have an indirect role in maintaining bacterial homeostasis by their effects on bacterial endosymbionts. This remains to be elucidated and will require the generation of recombinant proteins of PIXR 4 and 5. This is now included in the revised manuscript.

The Western Blot results shown in the Figure 2D and 2E detect the PIXR in nymphal midgut homogenate as a clear double band. To my opinion, it may reflect the presence of at least two closely related PIXR isoforms rather than the suggested post-translational modification of one protein. In order to unambiguously answer this question, another PIXR-RNAi silencing experiment in nymphs should be performed with following Western blot analysis of nymphal midgut homogenates RNAi. In case this experiment demonstrate two different PIXRs in nymphal midgut, than the redundancy may explain the lack of expected phenotypes.

We thank the reviewer for directing us to perform experiments to clarify that the doublet band is PIXR 1 and 2 (please note the detailed clarification above). This is now included in the revised manuscript.

3. In testing the potential antimicrobial activity of rPIXR against various bacteria (Supplemental Fig. 2), no positive control with any specific antibacterial peptide (defensin, cecropin or similar) or at least antibiotics was used. It is therefore impossible to judge whether the grow assays worked as performed.

*We agree and have now performed the suggested positive controls (Defensin for gram positive bacteria, Cecropin for gram negative bacteria and additionally gentamicin for both (Supplemental Fig 3). We have also assessed *Borrelia* growth in the presence of the antibiotic and the two antimicrobial peptides (Supplemental Fig 3).*

4. The results obtained by electron microscopy shown in Figure 4 A, B are at first look not much convincing. I realize that it is impossible to make a judgment without examining and comparing dozens of images of midgut sections from the control and experimental group. Therefore accepting this evidence and interpretation of the obtained results can be based only on the high credibility of the authors' team. A timepoint of 48h for EM and metabolome examination of nymphal midguts selected by the authors was reasoned to avoid the excessive contamination with the blood meal. On the other hand, it should be taken into consideration that at this time point, the experimental group may comprise a mixture of nymphs at different feeding progress with no chance to distinguish between male and female nymphs.

We agree that at this time points we cannot distinguish males and females. However, this is a random sampling of ticks from control and experimental groups over at least three replicate experiments. Routine sampling of repleted ticks in different experiments separates ticks into two roughly equal numbers of males and females (heavy, > 3.5 mg in weight representing females and light, < 3.5 mg representing males) (Figure 3A and (Schuijt et al., 2011)). Therefore, we expect that the electron microscopic examination of randomly sampled ticks should have accounted for an unbiased inclusion of both genders in both groups to similar extents.

How does the biofilm look in unfed ticks, is it formed at all?

This is indeed an interesting point. We have performed scanning electron microscopic visualization of unfed tick guts and observe no biofilm like structures suggesting that biofilms are likely increased during feeding (Figure 4C). We speculate that tick-associated bacteria tend to form biofilms as a potential strategy to sustain survival in the tick gut and to prevent from being excreted out during feeding.

5. A scapularisin type of defensin has been identified by subtractive hybridization to be upregulated in Bb-infected ticks. This is an interesting finding worth of further investigation. However, the authors get over this with a statement (lines 108-110) that the "scapularisin peptide has not been detected" with a reference to the review article by Sonenshine and Hynes, 2008. To my knowledge, there is no information about this particular scapularisin in this review. Instead the authors may better assign the identified scapularisin to one of 25 scapularisins described in I. scapularis (Wang & Zhu, 2011, Devel. Comp. Immunol, 351128-34).

We have now corrected this error in the revised manuscript as suggested and discussed the work by Wang and Zhu et al (Wang and Zhu, 2011).

Minor points (typos):

line 113 "Pediculis" change to "Pediculus"

line 124 "IXodes" change to "Ixodes"

line 130-131 "larvae that fed on clean or B. burgdorferi -infected larvae" change to "larvae that fed on clean or B. burgdorferi-infected mice"

line 473 "Syber Green" change to "Sybr Green"

line 576 "S. aureus" change to "S. aureus"

line 577 "P. aeruginosa" change to "P. aeruginosa", "B. casei" change to "B. casei"

We have corrected all the typos and done a careful spell check of the manuscript.

Reviewer #3 (Remarks to the Author):

What are the major claims of the paper?

Following on previous work by these authors that demonstrated that tick gut microbial composition influenced *Borrelia burgdorferi* (the causative agent of lyme disease) colonization, this study provides compelling evidence for the role of a protein with a Reeler domain (PIXR) in the facilitation of colonization of ticks by *B. burgdorferi*. They provide evidence to show that PIXR, is a secreted gut protein and is upregulated in the presence of *B. burgdorferi* and that when PIXR is inactivated, using both RNAi and by exposing to anti-PIXR antibodies, that *B. burgdorferi* is less effective at colonizing ticks. The mechanisms by which PIXR impacts *B. burgdorferi* colonization was investigated and a number of potential direct and indirect effects were observed and/or proposed, including impacts on Peritrophic Membrane structure, bacterial biofilm formation (possibly in a Gram-status specific manner), related impacts on gut microbiome composition and metabolites.

Are they novel and will they be of interest to others in the community and the wider field?

This study also expands on the growing body of literature related to Reeler domain proteins (such as Noduler) that play a key role in invertebrate response to microbial infection/colonization. In this case however the protein does not seem to have a bactericidal function, rather it appears to exert selection on the microbiome, reducing biofilm formation/expansion/ of Gram positive bacteria in a gut microbiome that is predominated by Gram negative bacteria. The selective nature of this interaction is fascinating and no doubt underlain by complex molecular recognition processes that are not discussed (nor within the scope of this study). Given the homology of PIXR within many other invertebrates, this study should be of broad appeal to many working in vector borne diseases, invertebrate ecology, biocontrol and host-microbe interactions in general.

If the conclusions are not original, it would be helpful if you could provide relevant references.

I believe the conclusions are original and in most cases supported by the data presented.

Is the work convincing, and if not, what further evidence would be required to strengthen the conclusions?

I had some difficulty in navigating the manuscript, although it is generally well written the results and discussion intertwine both the direct and indirect mechanisms by which PIXR could exert the observed phenotype. The manuscript also investigates PIXR impacts at both larval and nymph stage and in several sections it is not clear which growth stage is referred to – this is especially important given the timing of the development *Borrelia*-tick associations in nature (i.e. occurring at larval and not nymph stage). For this reason I think the manuscript would benefit from a summary figure of a conceptual model that highlights the potential mechanisms and evidence from both

stages. For example the authors describe alterations to the Peritrophic Membrane as observed using electron microscopy and Schiff staining, as a Reeler domain protein it is very possible that PIXR is involved in regulating the composition of the PM (in addition to bacterial EPS) and that this impacts colonization by B. burgdorferi. And while I tend to agree with the line of reasoning that the impact of PIXR is through selective suppression of Gram positive growth in biofilms, the authors previous work demonstrated that dysbiotic gut microbiota resulted in significant changes in expression of the STAT transcription factor – in this case STAT expression was not altered when PIXR was presumably abrogated through feeding on fed on PIXR-immunized mice. Other responses to dysbiotic gut microbiota observed in the previous study but not observed here include the decrease in peritrophin (component of the PM). This suggests that two or more separate mechanisms are at play. So while the combination of multiple lines of evidence makes this a very strong manuscript, I still think it would have benefited from an experiment to de-couple the direct and indirect effects of PIXR. For example, like in the authors previous manuscript, a suggestion would be to use an antibiotic treatment. In this case to suppress Gram-positive bacteria in the ticks fed on PIXR-immunized mice that showed increased Brevibacter – if the mechanism promoting B. burgdorferi colonization is related to repressing Gram-positive biofilm growth, then using Gram-positive specific antibiotics in this case would confirm this by reducing Brevibacter and the other Gram-positives and increasing B. burgdorferi in the ticks fed on PIXR-immunized mice.

We have now revised the manuscript to ensure a more cogent text and also included a graphical abstract highlighting the findings reported in this manuscript (Figure 9). Gram-positive bacteria-specific antibiotics that would selectively kill gram positive and not B. burgdorferi N40 are not available. Attempts to inject defensin an antibacterial peptide that readily targets gram-positive bacteria without compromising B. burgdorferi as seen in Supplemental Fig 3 did not provide reduction of gram-positive bacteria due potentially to the low amounts that we could inject and likely degradation of the injected defensin during feeding. Hence this is not included in the manuscript.

While the majority of the manuscript is string and evidence based, the connection to the metabolite composition and its role in serving as a cue for B. burgdorferi colonization is the weakest link. I admire the lengths the authors went to, to address this important question, but it was not nailed down in anyway and a large part of the discussion and the overall conclusions are based on what remains speculation. I am sure they are following this up in current work but those claims need to be tempered until the conclusive data are acquired.

We agree, and have now toned the discussion on metabolites significantly and have stated that this remains a speculation that requires further detailed analysis. As a curiosity, how does the gut microbiome of laboratory raised ticks compare to those in nature – obviously this would be important given the hypothesis here that the molecular cues for B. burgdorferi colonization are derived from the gut microbial metabolites. I know this has been addressed in previous manuscripts including by these authors but a discussion of this would be valuable.

We agree that this is an important extension of the described work and we have now provided a Discussion on this based on data available in literature.

Specific comments:

L330: What does gut genes refer to? Host genes or host and microbiome? Suggest rewording.

We have now referred to tick gut genes or B. burgdorferi genes expressed in the tick gut to clearly distinguish the reference to gut genes.

L331: I would describe the microbiome as more than a “new correlate”.

We agree and have now removed the word new correlate and describe it as “tick-associated microbiota that modulate tick-spirochete interactions”.

L369-373: There is so much other worthy detail that this connection to copper being toxic and Cupriavidus purifying the environment is just speculation and not worth including in my opinion without data. Also Cupriavidus is not ‘unique’ in that aspect.

We agree and we have now removed this speculation. Further, Cupriavidus is a minor component (<1 %) of the microbiome and was removed from this revised analysis.

L389: C-di-GMP upregulation is interesting for many reason, including its connection to EPS production in response to nitric oxide – that could be an important mechanism in B. burgdorferi establishment – maybe NO is necessary?

This is an interesting point, as there is increasing evidence that biofilm formation is likely regulated by increased nitric oxide potentially by increasing cyclic-di-GMP (Arora et al., 2015; Plate and Marletta, 2012) and this would also impact B. burgdorferi gene expression (Samuels, 2011). Bacteria can also generate NO via bacterial nitric oxide synthase (Crane et al., 2010) and whether increased gram-positive bacterial biofilms might impact NO levels in the gut has not been assessed in this study. Targeted metabolome studies would be required to address this possibility and is now noted in the Discussion.

L393-398: The metabolite data is still preliminary without identification – all this says is that metabolites are different, and there are some upregulated Borrelia genes that have been related to tick colonization (but that have not been shown to be essential for that process yet) but there is no way to assign any sort of causality at this point. Again I’m sure the authors are going there but the evidence in this paper does not support this so I suggest they ease off on the length of discussion around this topic and the emphasis in the abstract/discussion.

We agree and have now clarified the metabolome changes could potentially influence the B. burgdorferi transcriptome, but remains to be addressed in further detail.

Literature Cited.

Abraham, N.M., Liu, L., Jutras, B.L., Yadav, A.K., Narasimhan, S., Gopalakrishnan, V., Ansari, J.M., Jefferson, K.K., Cava, F., Jacobs-Wagner, C., et al. (2017). Pathogen-mediated manipulation of arthropod microbiota to promote infection. *Proceedings of the National Academy of Sciences of the United States of America* 114, E781-E790.

Andreotti, R., Perez de Leon, A.A., Dowd, S.E., Guerrero, F.D., Bendele, K.G., and Scoles, G.A. (2011). Assessment of bacterial diversity in the cattle tick *Rhipicephalus (Boophilus) microplus* through tag-encoded pyrosequencing. *BMC microbiology* 11, 6.

Arora, D.P., Hossain, S., Xu, Y., and Boon, E.M. (2015). Nitric Oxide Regulation of Bacterial Biofilms. *Biochemistry* 54, 3717-3728.

Budachetri, K., Browning, R.E., Adamson, S.W., Dowd, S.E., Chao, C.C., Ching, W.M., and Karim, S. (2014). An insight into the microbiome of the *Amblyomma maculatum* (Acari: Ixodidae). *Journal of medical entomology* 51, 119-129.

Budachetri, K., Gaillard, D., Williams, J., Mukherjee, N., and Karim, S. (2016). A snapshot of the microbiome of *Amblyomma tuberculatum* ticks infesting the gopher tortoise, an endangered species. *Ticks and tick-borne diseases* 7, 1225-1229.

Bylund, J., Burgess, L.A., Cescutti, P., Ernst, R.K., and Speert, D.P. (2006). Exopolysaccharides from *Burkholderia cenocepacia* inhibit neutrophil chemotaxis and scavenge reactive oxygen species. *The Journal of biological chemistry* 281, 2526-2532.

Carpi, G., Cagnacci, F., Wittekindt, N.E., Zhao, F., Qi, J., Tomsho, L.P., Drautz, D.I., Rizzoli, A., and Schuster, S.C. (2011). Metagenomic profile of the bacterial communities associated with *Ixodes ricinus* ticks. *PloS one* 6, e25604.

Chou, S., Daugherty, M.D., Peterson, S.B., Biboy, J., Yang, Y., Jutras, B.L., Fritz-Laylin, L.K., Ferrin, M.A., Harding, B.N., Jacobs-Wagner, C., et al. (2014). Transferred interbacterial antagonism genes augment eukaryotic innate immune function. *Nature*.

Clayton, K.A., Gall, C.A., Mason, K.L., Scoles, G.A., and Brayton, K.A. (2015). The characterization and manipulation of the bacterial microbiome of the Rocky Mountain wood tick, *Dermacentor andersoni*. *Parasites & vectors* 8, 632.

Crane, B.R., Sudhamsu, J., and Patel, B.A. (2010). Bacterial nitric oxide synthases. *Annual review of biochemistry* 79, 445-470.

Cywes-Bentley, C., Skurnik, D., Zaidi, T., Roux, D., Deoliveira, R.B., Garrett, W.S., Lu, X., O'Malley, J., Kinzel, K., Zaidi, T., et al. (2013). Antibody to a conserved antigenic target is protective against diverse prokaryotic and eukaryotic pathogens. *Proceedings of the National Academy of Sciences of the United States of America* 110, E2209-2218.

Delaney, J.R., Stoven, S., Uvell, H., Anderson, K.V., Engstrom, Y., and Mlodzik, M. (2006). Cooperative control of *Drosophila* immune responses by the JNK and NF-kappaB signaling pathways. *The EMBO journal* 25, 3068-3077.

Gall, C.A., Reif, K.E., Scoles, G.A., Mason, K.L., Mousel, M., Noh, S.M., and Brayton, K.A. (2016). The bacterial microbiome of *Dermacentor andersoni* ticks influences pathogen susceptibility. *The ISME journal* 10, 1846-1855.

Hammer, B., Moter, A., Kahl, O., Alberti, G., and Gobel, U.B. (2001). Visualization of *Borrelia burgdorferi sensu lato* by fluorescence in situ hybridization (FISH) on whole-body sections of *Ixodes ricinus* ticks and gerbil skin biopsies. *Microbiology* 147, 1425-1436.

Jensen, E.T., Kharazmi, A., Garred, P., Kronborg, G., Fomsgaard, A., Mollnes, T.E., and Hoiby, N. (1993). Complement activation by *Pseudomonas aeruginosa* biofilms. *Microbial pathogenesis* 15, 377-388.

Khokha, M.K., Chung, C., Bustamante, E.L., Gaw, L.W., Trott, K.A., Yeh, J., Lim, N., Lin, J.C., Taverner, N., Amaya, E., et al. (2002). Techniques and probes for the study of *Xenopus tropicalis* development. *Developmental dynamics : an official publication of the American Association of Anatomists* 225, 499-510.

Klyachko, O., Stein, B.D., Grindle, N., Clay, K., and Fuqua, C. (2007). Localization and visualization of a coxiella-type symbiont within the lone star tick, *Amblyomma americanum*. *Applied and environmental microbiology* 73, 6584-6594.

Kurilshikov, A., Livanova, N.N., Fomenko, N.V., Tupikin, A.E., Rar, V.A., Kabilov, M.R., Livanov, S.G., and Tikunova, N.V. (2015). Comparative Metagenomic Profiling of Symbiotic Bacterial Communities Associated with *Ixodes persulcatus*, *Ixodes pavlovskyi* and *Dermacentor reticulatus* Ticks. *PloS one* 10, e0131413.

Lemaitre, B. (2004). The road to Toll. *Nature reviews Immunology* 4, 521-527.

Menchaca, A.C., Visi, D.K., Strey, O.F., Teel, P.D., Kalinowski, K., Allen, M.S., and Williamson, P.C. (2013). Preliminary assessment of microbiome changes following blood-feeding and survivorship in the *Amblyomma americanum* nymph-to-adult transition using semiconductor sequencing. *PloS one* 8, e67129.

Murofushi, Y., Villena, J., Morie, K., Kanmani, P., Tohno, M., Shimazu, T., Aso, H., Suda, Y., Hashiguchi, K., Saito, T., et al. (2015). The toll-like receptor family protein RP105/MD1 complex is involved in the immunoregulatory effect of exopolysaccharides from *Lactobacillus plantarum* N14. *Molecular immunology* 64, 63-75.

Nakao, R., Abe, T., Nijhof, A.M., Yamamoto, S., Jongejan, F., Ikemura, T., and Sugimoto, C. (2013). A novel approach, based on BLSOMs (Batch Learning Self-Organizing Maps), to the microbiome analysis of ticks. *The ISME journal* 7, 1003-1015.

Narasimhan, S., Rajeevan, N., Liu, L., Zhao, Y.O., Heisig, J., Pan, J., Eppler-Epstein, R., Deponte, K., Fish, D., and Fikrig, E. (2014). Gut microbiota of the tick vector

Ixodes scapularis modulate colonization of the Lyme disease spirochete. *Cell host & microbe* 15, 58-71.

Palmer, W.J., and Jiggins, F.M. (2015). Comparative Genomics Reveals the Origins and Diversity of Arthropod Immune Systems. *Molecular biology and evolution* 32, 2111-2129.

Plate, L., and Marletta, M.A. (2012). Nitric oxide modulates bacterial biofilm formation through a multicomponent cyclic-di-GMP signaling network. *Molecular cell* 46, 449-460.

Salter, S.J., Cox, M.J., Turek, E.M., Calus, S.T., Cookson, W.O., Moffatt, M.F., Turner, P., Parkhill, J., Loman, N.J., and Walker, A.W. (2014). Reagent and laboratory contamination can critically impact sequence-based microbiome analyses. *BMC biology* 12, 87.

Samuels, D.S. (2011). Gene regulation in *Borrelia burgdorferi*. *Annual review of microbiology* 65, 479-499.

Schuijt, T.J., Narasimhan, S., Daffre, S., DePonte, K., Hovius, J.W., Van't Veer, C., van der Poll, T., Bakhtiari, K., Meijers, J.C., Boder, E.T., et al. (2011). Identification and characterization of *Ixodes scapularis* antigens that elicit tick immunity using yeast surface display. *PloS one* 6, e15926.

Smith, A.A., Navasa, N., Yang, X., Wilder, C.N., Buyuktanir, O., Marques, A., Anguita, J., and Pal, U. (2016). Cross-Species Interferon Signaling Boosts Microbicidal Activity within the Tick Vector. *Cell host & microbe* 20, 91-98.

Smith, A.A., and Pal, U. (2014). Immunity-related genes in *Ixodes scapularis*--perspectives from genome information. *Frontiers in cellular and infection microbiology* 4, 116.

Trout Fryxell, R.T., and DeBruyn, J.M. (2016). The Microbiome of Ehrlichia-Infected and Uninfected Lone Star Ticks (*Amblyomma americanum*). *PloS one* 11, e0146651.
Van Treuren, W., Ponnusamy, L., Brinkerhoff, R.J., Gonzalez, A., Parobek, C.M., Juliano, J.J., Andreadis, T.G., Falco, R.C., Ziegler, L.B., Hathaway, N., et al. (2015). Variation in the Microbiota of *Ixodes* Ticks with Regard to Geography, Species, and Sex. *Applied and environmental microbiology* 81, 6200-6209.

Wang, Y., and Zhu, S. (2011). The defensin gene family expansion in the tick *Ixodes scapularis*. *Developmental and comparative immunology* 35, 1128-1134.

Watters, C., Fleming, D., Bishop, D., and Rumbaugh, K.P. (2016). Host Responses to Biofilm. *Progress in molecular biology and translational science* 142, 193-239.

Reviewers' comments:

Reviewer #1 (Remarks to the Author):

Narasimhan et al have satisfactorily addressed my major concerns with the initial submission.

Remaining comment:

- The authors must delineate western blots. For instance, those lanes in Fig 2D and 2E representing a single original gel should be boxed. I am concerned that lane 1 of the α -salp25D loading control blot for 2E appears to be inappropriately spliced from another gel, which is misleading.

Reviewer #2 (Remarks to the Author):

The revised manuscript by Narasimhan et al, entitled "PIXR, a secreted *Ixodes scapularis* protein, modulates the gut milieu in favor of *Borrelia burgdorferi* colonization of the tick" is fundamentally improved compared to the original submission and meet most of my comments and suggestions raised in my previous review. Also I am completely satisfied with the authors' responses in the rebuttals as they are quite informative and explanatory.

Major point:

In my opinion, there remains one major point which needs to be resolved before the revised manuscript can be accepted for publication, namely the complete sequence of PIXR2.

All the experiments shown by the authors point to the fact that the described effects of PIXR is actually the simultaneous effect of two closely related and similarly expressed proteins PIXR1 and PIXR2. The sequence alignment shown in Figure 1C clearly indicates that the sequence of PIXR2 (geneISCW017733) is not complete with an obvious C-terminal gap. Therefore all the speculations about differences in molecular mass differences or the effect of possible glycosylation between PIXR1 and PIXR2 (lines 154-158) seem to me to be rather inappropriate. Instead, the authors should clone the *pixr2* gene from the midgut cDNA, deposit its correct sequence in the GenBank and include the correct sequence of PIXR2 in the Fig 1A and 1C. My search in the midgut sequences from the closely related *Ixodes ricinus* suggests that the closest ortholog of *pixr2* is most likely the transcript Ir-108734 (GenBank JAP74734). Unfortunately, the sequence of *I. scapularis* *pixr1* (Acc. No KY629420) is still not available from the GenBank which prevented me to verify whether or not the *I. ricinus* sequence displayed in the Figure 1A is indeed the closest ortholog of *pixr1* (please, provide all the accession numbers to Figure 1A in the legend).

Once having the complete sequence of *pixr2*, the authors should align nucleotide sequences of *pixr1* and *pixr2* that may aid to the explanation why *pixr1* dsRNA silenced expression of both genes.

Minor points:

1. Manuscript line 90 and lines 937-939 and supplemental table 1 line - Wrong citation 17. This reference does not cite a paper on tick asparaginyl endopeptidase (legumain) as intended but a non-related paper on tick serine carboxypeptidase.
2. Figure 3C-I: description of lines 1-4 (ds *gfp*?) and 5-8 (ds *pixr*?) is missing
3. Figure 4 B-iii. The panel B-iii (also referred to in the text) is described in the legend as panel C. Match the legend with the Figure.

4. Supplemental Table 2: Correct the ISCW Nos. for pixr2, 3 and 4 (remove one null)

Reviewer #3 (Remarks to the Author):

The author's have addressed my most significant concerns where practical to do so.

Response to Reviewers comments

We are grateful for the careful review of the revised manuscript and we have incorporated the additional changes requested by the reviewers to further clarify the observations. Our point-by-point response (in italics) to their specific comments (underlined) are provided below.

Reviewer #1 (Remarks to the Author):

Reviewer #1 (Remarks to the Author):

Narasimhan et al have satisfactorily addressed my major concerns with the initial submission.

Remaining comment:

The authors must delineate western blots. For instance, those lanes in Fig 2D and 2E representing a single original gel should be boxed. I am concerned that lane 1 of the a-salp25D loading control blot for 2E appears to be inappropriately spliced from another gel, which is misleading.

As permitted by the journal, the only processing of the blots involved brightness adjustment of the blots across the entire image in the interest of clarity. The data presented are indeed representative of the original blots. Original images of the blots including the markers are now provided as a Supplemental Figure 8 and stated in the Data availability section of the revised manuscript. The marker bands in Supplementary Figure 8A appear white due to saturating signal intensity and this is noted in the legend for clarity.

We have now boxed separate blots in Fig 2D and 2E that shows replicate blots probed with different polyclonal antisera.

We confirm that lane 1 in Fig 2E does not represent an irrelevant sample. Lane 1 represents unfed whole larval extract in comparison to unfed nymphal guts of lanes 2 and 3. Although comparable amounts of unfed protein extracts were loaded larval extract represents whole larvae and potentially also includes other tick tissues including salivary glands and might account for the increased intensity for Salp25D.

We have now repeated the immunoblot to clarify the images to include all samples on the same run as requested, and the new blot is provided in Fig 2E. We have also probed the blot with naïve/pre-immune sera to additionally highlight specific reactivity to PIXR, and demonstrate equivalent loading amounts. Salp25D is used as a control to also demonstrate the integrity of the samples. While traditionally actin serves as a loading control, detection of tick actin is confounded by the presence of host blood and actin therein and hence not used in these experiments involving fed ticks. We have also provided the full western blot images of Fig 3C including the markers in Supplement Figure 8.

Reviewer #2 (Remarks to the Author):

The revised manuscript by Narasimhan et al, entitled “PIXR, a secreted Ixodes scapularis protein, modulates the gut milieu in favor of Borrelia burgdorferi colonization of the tick” is fundamentally improved compared to the original submission and meet most of my comments and suggestions raised in my previous review. Also I am completely satisfied with the authors’ responses in the rebuttals as they are quite informative and explanatory.

Major point:

In my opinion, there remains one major point which needs to be resolved before the revised manuscript can be accepted for publication, namely the complete sequence of PIXR2. All the experiments shown by the authors point to the fact that the described effects of PIXR is actually the simultaneous effect of two closely related and similarly expressed proteins PIXR1 and PIXR2. The sequence alignment shown in Figure 1C clearly indicates that the sequence of PIXR2 (geneISCW017733) is not complete with an obvious C-terminal gap. Therefore all the speculations about differences in molecular mass differences or the effect of possible glycosylation between PIXR1 and PIXR2 (lines 154-158) seem to me to be rather inappropriate. Instead, the authors should clone the pixr2 gene from the midgut cDNA, deposit its correct sequence in the GenBank and include the correct sequence of PIXR2 in the Fig 1A and 1C. My search in the midgut sequences from the closely related Ixodes ricinus suggests that the closest ortholog of pixr2 is most likely the transcript Ir-108734 (GenBank JAP74734). Unfortunately, the sequence of I. scapularis pixr1 (Acc. No KY629420) is still not available from the GenBank which prevented me to verify whether or not the I. ricinus sequence displayed in the Figure 1A is indeed the closest ortholog of pixr1 (please, provide all the accession numbers to Figure 1A in the legend).

We agree with the reviewer and as suggested we have now cloned and sequenced the full-length PIXR2 transcript represented in the tick gut. The full-length gene encodes for a ~15.4 kDa protein and is now consistent with the mobility observed by gel electrophoresis. This is now acknowledged in the revised text. PIXR2 sequence is now deposited in the public database (GenBank #KY865270). PIXR sequence (GenBank #KY629420) is now available in the database. We have also provided the nucleotide and protein sequence of PIXR and PIXR2 as a word document to facilitate the review process –should there be any delays in the release of the GenBank submitted data. We have now included all the accession numbers of genes shown in Fig 1A.

Once having the complete sequence of pixr2, the authors should align nucleotide sequences of pixr1 and pixr 2 that may aid to the explanation why pixr1 dsRNA silenced expression of both genes.

As suggested we have now aligned the sequences of pixr and pixr2 (Supplemental Fig 7) to provide further supporting evidence that pixr1 dsRNA would also silence pixr2.

Minor points:

1. Manuscript line 90 and lines 937-939 and supplemental table 1 line - Wrong citation 17. This reference does not cite a paper on tick asparaginyl endopeptidase (legumain) as intended but a non-related paper on tick serine carboxypeptidase.

We have now corrected this error and included the correct citation

2. Figure 3C-I: description of lines 1-4 (ds gfp?) and 5-8 (ds pixr?) is missing

We apologize for this omission-this has been included now.

3. Figure 4 B-iii. The panel B-iii (also referred to in the text) is described in the legend as panel C. Match the legend with the Figure.

This error is now corrected.

4. Supplemental Table 2: Correct the ISCW Nos. for pixr2, 3 and 4 (remove one null)

The correct gene numbers are now provided.

Reviewer #3 (Remarks to the Author):

The author's have addressed my most significant concerns where practical to do so.

REVIEWERS' COMMENTS:

Reviewer #2 (Remarks to the Author):

I am pleased to see that the authors performed the cloning and sequencing of *pixr2* and added appropriately the related data into the manuscript and Supplemental Figure 7. The authors also corrected all the errors I was able to identify in the previous version. Now, I only look forward seeing this manuscript published in Nature Communication.